# WNK1-dependent water influx is required for CD4+ T cell activation and T cell-dependent antibody responses

Joshua Biggs O'May [1], Lesley Vanes[1], Leonard L. de Boer [1,2,3], David A. Lewis [1], Harald Hartweger [1,4], Simone Kunzelmann [1], Darryl Hayward[1,5], Miriam Llorian [1], Robert Köchl[1,6] & Victor L. J. Tybulewicz [1] ✉

Signaling from the T cell antigen receptor (TCR) on CD4+ T cells plays a critical role in adaptive immune responses by inducing T cell activation, proliferation, and differentiation. Here we demonstrate that WNK1, a kinase implicated in osmoregulation in the kidney, is required in T cells to support T-dependent antibody responses. We show that the canonical WNK1-OXSR1-STK39 kinase signaling pathway is required for TCR signaling in CD4+ T cells, their subsequent entry into the cell cycle, and suppression of the ATR-mediated G2/M cell cycle checkpoint. We show that the WNK1 pathway regulates ion influx leading to water influx, potentially through AQP3, and that water influx is required for TCR-induced signaling and cell cycle entry. Thus, TCR signaling via WNK1, OXSR1, STK39 and AQP3 leads to water entry that is essential for CD4+ T cell proliferation and hence T cell-dependent antibody responses.

T cell activation is a critical event in the initiation of T cell-dependent (TD) immune responses. To generate TD antibody responses, naive CD4+ T cells circulate through lymphoid organs scanning peptides presented on MHC class II molecules (pMHCII) on the surface of antigen-presenting cells (APC)[1]. Simultaneous binding of cognate pMHCII complexes to the T cell antigen receptor (TCR) and ligation of costimulatory receptors such as CD28 results in biochemical signals leading to LFA-1 integrin-mediated adhesion of the T cell to the APC, and subsequent activation of the T cell characterized by changes in gene expression, cytokine secretion, cell division and differentiation into effector T cells such as T follicular helper ($T_{FH}$) cells[2]. $T_{FH}$ cells provide help to antigen-specific B cells, supporting their continued activation, proliferation, and differentiation into antibody-secreting plasma cells, thereby generating an antibody response.

WNK1 has been studied most extensively in the kidney where it acts in epithelial cells of the distal tubules to regulate uptake of ions from urine into the blood stream[3,4]. WNK1 phosphorylates and activates OXSR1 and STK39, two related kinases, which in turn phosphorylate members of the electroneutral SLC12A-family of ion cotransporters, leading to the net influx of $Na^+$, $K^+$ and $Cl^-$ ions[5–8]. This WNK-induced ion influx in turn leads to passive water influx, which underlies the role of WNK1 in regulating cell volume[4,9].

Our previous studies in CD4+ T cells showed that signaling from the TCR and from the CCR7 chemokine receptor acting via phosphatidylinositol 3-kinase leads to rapid activation of WNK1[10]. Furthermore, we found that WNK1 regulates both adhesion and migration of T cells. WNK1 is a negative regulator of TCR- and CCR7-induced adhesion through the LFA-1 integrin. Conversely, WNK1 is a positive regulator of CCR7-induced T cell migration. CCL21 binding to CCR7 results in signaling via WNK1, OXSR1, STK39 and SLC12A2 which is required for migration. In view of the importance of adhesion and migration for T cell activation, we hypothesized that WNK1 may have a critical role in T cells during TD immune responses.

[1]The Francis Crick Institute, London NW1 1AT, UK. [2]Imperial College, London W12 0NN, UK. [3]Present address: Science for Life Laboratory, Department of Women's and Children's Health, Karolinska Institute, Box 1031, SE-171 21 Solna, Sweden. [4]Present address: Laboratory of Molecular Immunology, The Rockefeller University, 10065 New York, NY, USA. [5]Present address: GSK, Stevenage SG1 2NY, UK. [6]Present address: Kings College London, London SE1 9RT, UK. ✉e-mail: Victor.T@crick.ac.uk

Here we show that WNK1-deficient T cells are unable to support a TD antibody response. Surprisingly, we find that in addition to its role in T cell adhesion and migration, WNK1 is required for TCR/CD28-induced activation. We show that in CD4+ T cells, TCR signaling via WNK1, OXSR1 and STK39 leads to ion influx and that subsequent water influx, in part through AQP3, is required for TCR signaling, T cell proliferation and hence TD antibody responses. Moreover, given the broad expression of WNK1, WNK1-dependent water influx may be a common feature of mitogenic pathways in many cell types, both within the immune system and beyond.

## Results

### WNK1 is essential for the generation of T_FH cells and class-switched antibody responses

To investigate if WNK1 is required in T cells for TD immune responses, we inducibly inactivated the *Wnk1* gene in mature T cells, since WNK1 is required for T cell development in the thymus[11]. Using mice with a loxP-flanked allele of *Wnk1* (*Wnk1*^fl)[12] crossed to mice with a loss of function *Wnk1* allele (*Wnk1*^-)[10] and mice with a tamoxifen-inducible Cre recombinase expressed from the ROSA26 locus (*ROSA26*^CreERT2, RCE)[13], we generated *Wnk1*^fl/-RCE mice and control *Wnk1*^fl/+RCE mice. Bone marrow from these two mouse strains was used to reconstitute irradiated RAG1-deficient mice, and 8 weeks later treatment with tamoxifen resulted in the generation of WNK1-deficient (*Wnk1*^-/-RCE) and control WNK1-expressing (*Wnk1*^+/-RCE) T cells (Supplementary Fig. 1a). *Wnk1* was efficiently deleted in CD4+ T cells from these chimeras following tamoxifen administration (Supplementary Fig. 1b). Importantly, loss of WNK1 did not result in compensatory increase in expression of *Wnk2*, *Wnk3*, or *Wnk4* as measured by RNA sequencing (Supplementary Fig. 1c). Thus, *Wnk1*^-/-RCE CD4+ T cells lack expression of wild-type forms of all *Wnk*-family genes. Further analysis of other WNK1 pathway proteins showed that loss of WNK1 resulted in small changes in expression of *Oxsr1*, *Stk39* and *Slc12a*-family genes (Supplementary Fig. 1d, e). Using mixed bone marrow chimeras in which *Wnk1* was deleted from all αβT cells, while other cell types retained *Wnk1* expression, we found that following immunization with NP-CGG in alum, the mice failed to generate NP-specific IgG1 antibodies, a hallmark TD response (Fig. 1a, b). Thus, WNK1 is required in αβT cells for TD antibody responses.

To identify the WNK1-dependent processes contributing to TD antibody responses, we generated *Wnk1*^fl/+RCE OT-II and *Wnk1*^fl/-RCE OT-II bone marrow chimeras which express a TCR (OT-II) specific for an ovalbumin peptide (OVA_323-339) presented on the I-A^b MHC class II molecule[14]. We treated the chimeras with tamoxifen, isolated WNK1-expressing and WNK1-deficient CD4+ OT-II T cells and transferred these into congenically-marked wild-type (WT) mice which were immunized with ovalbumin (Fig. 1c). Flow cytometric analysis showed that WNK1-deficient OT-II T cells were severely impaired in proliferation at day 3, and at day 7 many fewer WNK1-deficient OT-II T cells were recovered compared to controls, and in particular there were very few T_FH cells (Fig. 1d-j). The almost complete absence of T_FH cells likely explains the impaired TD antibody response in mice with WNK1-deficient T cells.

### WNK1, OXSR1 and STK39 are required for TCR-induced proliferation of CD4+ T cells

This defect could be due to altered T cell adhesion or migration. Alternatively, since TCR signaling activates WNK1, the kinase could play a direct role in TCR-induced activation. To study this possibility, we investigated whether the activation defect of WNK1-deficient T cells could be recapitulated in vitro. We cultured WNK1-expressing and WNK1-deficient CD4+ OT-II T cells with bone marrow-derived APCs and different doses of OVA_323-339 peptide (Supplementary Fig. 2a). We found that in the absence of WNK1, OT-II T cells were impaired in proliferation, similar to the phenotype observed in vivo (Fig. 2a, b). Since in this T cell:APC co-culture assay T cells need to adhere to APCs

and receive signals from multiple cell surface proteins in addition to the TCR, we simplified the in vitro analysis further by activating purified naive CD4+ T cells with plate-bound anti-CD3ε and anti-CD28 antibodies to stimulate the TCR and CD28 respectively (Supplementary Fig. 2b). Once again, we found that WNK1-deficient T cells were severely impaired in TCR/CD28-induced proliferation (Fig. 2c, d). This observation suggests that the requirement for WNK1 during CD4+ T cell activation is independent of its role in migration and integrin-mediated adhesion as T cells do not need to migrate or adhere to APCs in order to be activated in this experimental system. Our results show that WNK1 is required for TCR/CD28-induced proliferation, explaining the strongly impaired CD4+ T cell expansion in vivo during a TD immune response.

To evaluate the role of WNK1 kinase activity in this process, we analyzed TCR/CD28-induced activation of CD4+ T cells expressing a kinase-inactive allele of WNK1 (*Wnk1*^D368A)[10], a mutation of a single amino acid that has been shown to eliminate kinase activity (Supplementary Fig. 2b)[7]. The mutation did not affect the expression of *Wnk1* mRNA (Supplementary Fig. 2c) but eliminated TCR-induced WNK1 activity as assessed by OXSR1-Ser325 phosphorylation[10]. Once again, T cells expressing WNK1-D368A showed strongly decreased cell division, similar to the phenotype of WNK1-deficient T cells (Fig. 2e, f). To address whether the phenotypic changes observed were due to adaptation of cells to the chronic absence of WNK1 activity, rather than a requirement for WNK1 during TCR stimulation, we made use of WNK463 (WNKi), a highly selective WNK kinase inhibitor[15]. Activation of WT CD4+ T cells in the presence of WNKi showed that acute inhibition of WNK1, the only WNK-family kinase expressed in T cells (Supplementary Fig. 1c), resulted once again in dramatically impaired proliferation upon TCR and CD28 co-stimulation (Supplementary Fig. 2d, Fig. 2g, h). Taken together, these results show that WNK1 kinase activity is required for TCR/CD28-induced proliferation of CD4+ T cells.

The best characterized WNK1 substrates are the related OXSR1 and STK39 kinases[5-8]. Stimulation of CD4+ T cells through the TCR and CD28 resulted in an increase in phosphorylation of OXSR1 on Ser325, a site phosphorylated by WNK1[7] (Supplementary Fig. 3a). Treatment with WNKi caused a strong reduction in this phosphorylation, demonstrating that WNK1 kinase activity is required both for TCR-CD28-induced and basal phosphorylation of OXSR1. Thus, we investigated if these proteins might be important for WNK1-dependent T cell activation. We again used mice with a loxP-flanked allele of *Oxsr1* (*Oxsr1*^fl), which could be inducibly deleted by tamoxifen treatment. To account for potential redundancy between OXSR1 and STK39, we crossed in the *Stk39*^T243A allele which expresses an STK39 mutant that cannot be phosphorylated and activated by WNK1[16]. We used bone marrow from *Oxsr1*^fl/fl*Stk39*^T243A/T243ARCE as well as control *Oxsr1*^+/+*Stk39*^+/+RCE mice to reconstitute irradiated RAG1-deficient mice (Supplementary Fig. 3b). OXSR1 was efficiently deleted in mature CD4+ T cells from these chimeras following tamoxifen administration (Supplementary Fig. 3c). *Oxsr1*^-/-*Stk39*^T243A/T243ARCE double mutant CD4+ T cells exhibited reduced TCR/CD28-induced proliferation, however, the decrease was smaller than that seen in the absence of WNK1 (Fig. 2i, j). Our results suggest that a WNK1-OXSR1-STK39 pathway is required for CD4+ T cell proliferation.

### The WNK1 pathway is required for progression through the cell cycle

To gain a better understanding of how the WNK1 pathway impacts on T cell proliferation, we evaluated how the cells progress through the cell cycle in the absence of WNK1. Ki67 expression is induced as cells progress from G0 to G1 phase of the cell cycle. Following TCR/CD28 stimulation, WNK1-deficient T cells upregulated Ki67 more slowly than control cells, indicating that their entrance into the cell cycle is delayed (Fig. 3a). Analysis of Ki67+ cells 3 days after the start of TCR/CD28 stimulation showed that an increased fraction of

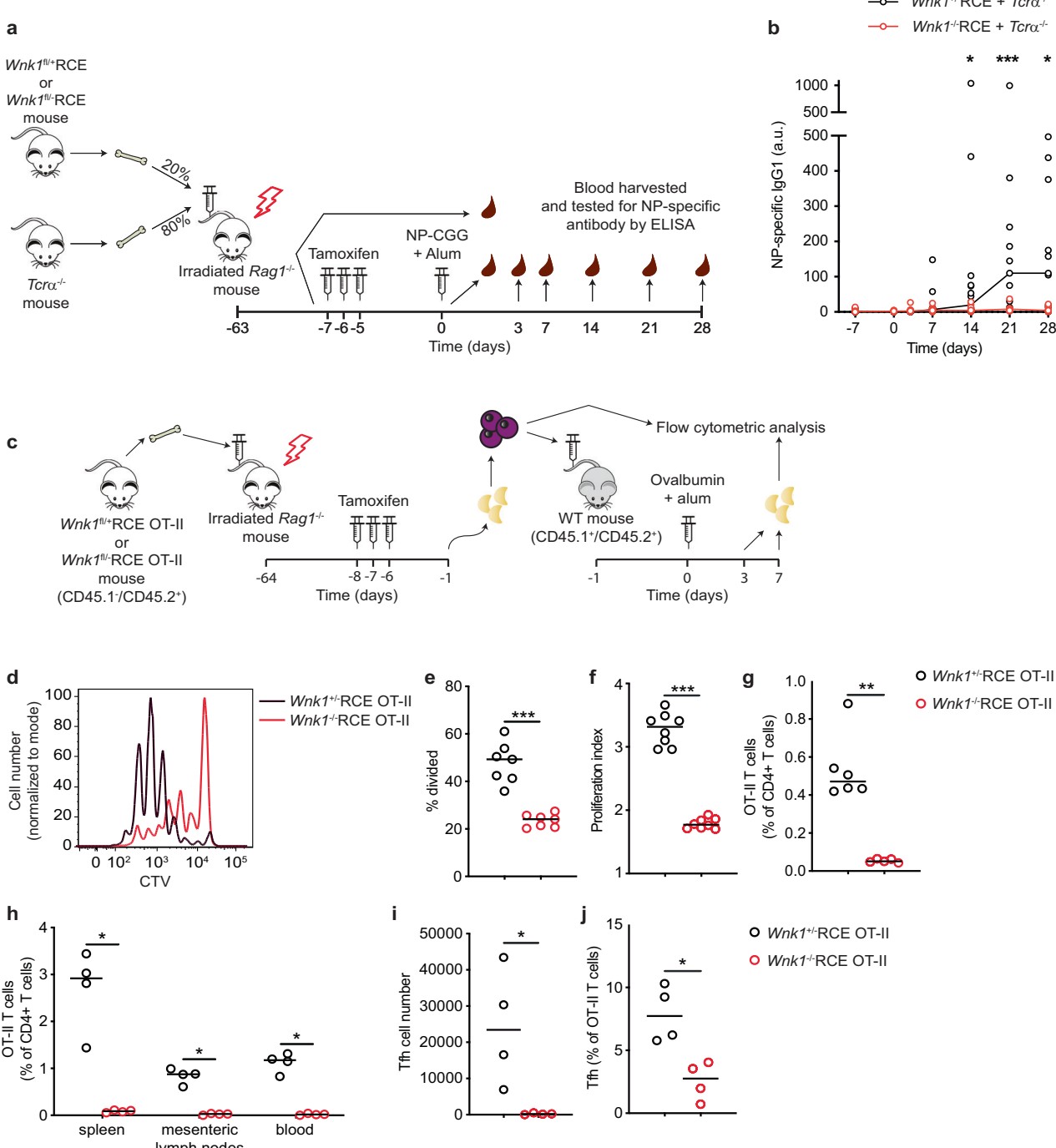

**Fig. 1 | WNK1 is essential for the generation of T$_{FH}$ cells and class-switched antibody responses. a, b** Irradiated RAG1-deficient mice were reconstituted with a mixture of bone marrow from TCRα-deficient (TCRα$^{-/-}$) mice and either *Wnk1*$^{fl/+}$RCE or *Wnk1*$^{fl/-}$RCE mice, treated with tamoxifen at least 8 weeks later to delete *Wnk1*$^{fl}$ alleles and immunized with NP-CGG in alum (**a**). NP-specific IgG1 antibodies in the serum quantified by ELISA at the time points indicated (**b**) (*Wnk1*$^{+/-}$RCE: d-7, $n = 18$; d0, 3, 7, $n = 17$; d14, $n = 16$; d21, 28, $n = 11$. *Wnk1*$^{-/-}$RCE d-7, $n = 16$; d0, 3, 7, 14, $n = 15$; d21, $n = 11$; d28, $n = 9$). Each point represents a different mouse, line indicates median. **c–j** Irradiated RAG1-deficient mice were reconstituted with bone marrow from *Wnk1*$^{fl/+}$RCE OT-II or *Wnk1*$^{fl/-}$RCE OT-II mice, treated with tamoxifen, and 7 d later naïve *Wnk1*$^{+/-}$ or *Wnk1*$^{-/-}$ CD4$^+$ OT-II T cells (CD45.1$^-$CD45.2$^+$) were isolated, labelled with CTV, adoptively transferred into WT (CD45.1$^+$CD45.2$^+$) mice that were subsequently immunized with OVA in alum, and 3 or 7 days later, lymph node cells were analyzed by flow cytometry (**c**). CTV dilution at day 3 (**d**) was used to calculate the percentage of cells that had divided at least once (**e**, $n = 7$) and the proliferation index (average number of divisions by dividing cells) (**f**, $n = 8$). CD4$^+$CD45.1$^-$CD45.2$^+$ OT-II T cells as a percentage of all CD4$^+$ cells in peripheral and mesenteric lymph nodes at day 3 (**g**, *Wnk1*$^{+/-}$RCE OT-II: $n = 6$, *Wnk1*$^{-/-}$RCE OT-II: $n = 5$), and in the indicated tissues at day 7 (**h**, $n = 4$). Splenic T$_{FH}$ (PD-1$^{hi}$CXCR5$^+$) cell numbers (**i**) and percentage of all OT-II T cells day 7 (**j**, $n = 4$). Each point represents a different mouse. Horizontal lines show median. Data are from 1 experiment representative of 2 (**g**) or 3 (**h–j**) experiments, or pooled from 2 independent experiments (**b, e, f**). a.u., arbitrary units. Statistical analysis with the 2-sided Mann-Whitney U test. *0.01 <p < 0.05; **0.001 <p < 0.01; ***0.0001 <p < 0.001. Source data are provided as a Source Data file.

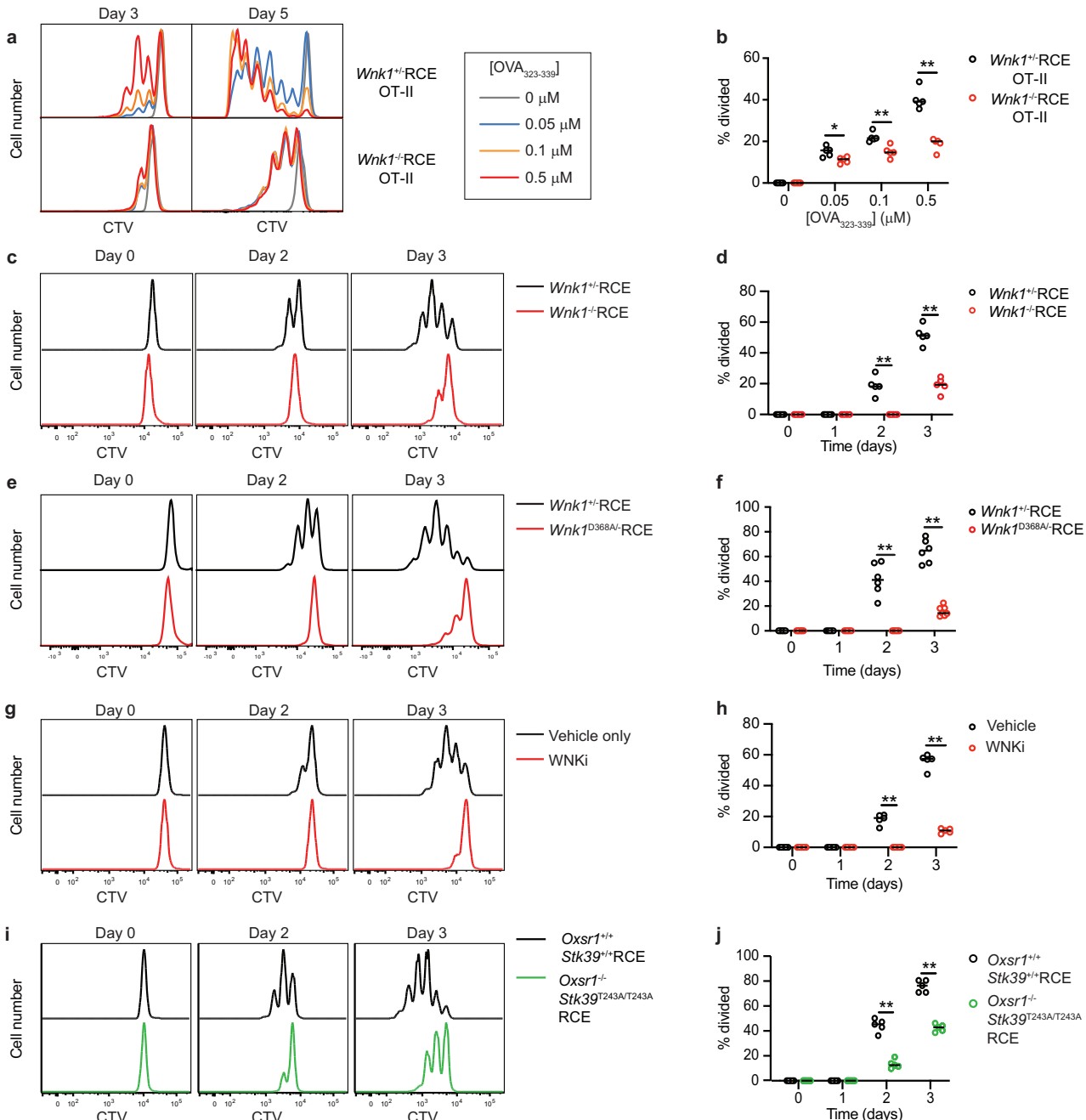

**Fig. 2 | WNK1, OXSR1, and STK39 are required for TCR-induced proliferation of CD4⁺ T cells. a, b** WNK1-expressing or -deficient CTV-labelled OT-II CD4⁺ T cells were co-cultured with bone marrow-derived APCs in the presence of indicated concentrations of OVA$_{323-339}$ peptide for 5 days. CTV dilution (**a**) was used to determine the percentage of cells that had divided at day 3 (**b**, $n = 5$). (**c**–**j**) Naïve CD4⁺ CTV-labelled T cells of the indicated genotypes (**c**–**f, i, j**) or treated with WNKi (**g, h**) were activated on plate-bound anti-CD3ε and anti-CD28 antibodies for indicated times. CTV dilution (**c, e, g, i**) was used to determine the percentage of cells

that had divided (**d, h, j**, $n = 5$; **f**, $Wnk1^{+/-}$RCE: d0, 2, 3, $n = 6$; d1, $n = 5$. $Wnk1^{D368A/-}$RCE: d0, 3, $n = 7$; d1, 2, $n = 6$). WNKi, WNK inhibitor (WNK463). Each point represents a different mouse; horizontal lines show median; data are from 1 experiment representative of 2 (**b, h**) or 3 (**d, j**) experiments, or pooled from 2 independent experiments (**f**). Statistical analysis with the 2-sided Mann-Whitney U test. *$0.01 < p < 0.05$; **$0.001 < p < 0.01$. Source data are provided as a Source Data file.

WNK1-deficient cells accumulated in the G2 phase relative to controls (Fig. 3b, c, Supplementary Fig. 4a), suggesting that their reduced rate of proliferation may also be due to slower progression through G2 or arrest at the G2/M checkpoint. Furthermore, the rate of incorporation of the nucleoside analog EdU was greatly reduced in WNK1-deficient T cells indicating that DNA replication and progression through S phase is also severely impaired in the absence of WNK1 (Fig. 3d, e).

In response to TCR/CD28 co-stimulation, $Oxsr1^{-/-}Stk39^{T243A/T243A}$RCE double mutant T cells also upregulated Ki67 more slowly than control

cells, accumulated in S and G2 phases of the cell cycle and had reduced rates of EdU incorporation (Fig. 3f-j, Supplementary Fig. 4b). Thus WNK1, OXSR1 and STK39 are required for CD4⁺ T cells to efficiently exit G0 and progress through S and G2.

## The WNK1 pathway is required for early TCR signaling
The defect in G0 exit in TCR/CD28-stimulated WNK1-deficient T cells could be caused by alterations in early signaling events triggered following engagement of TCR and CD28, which provide the mitogenic

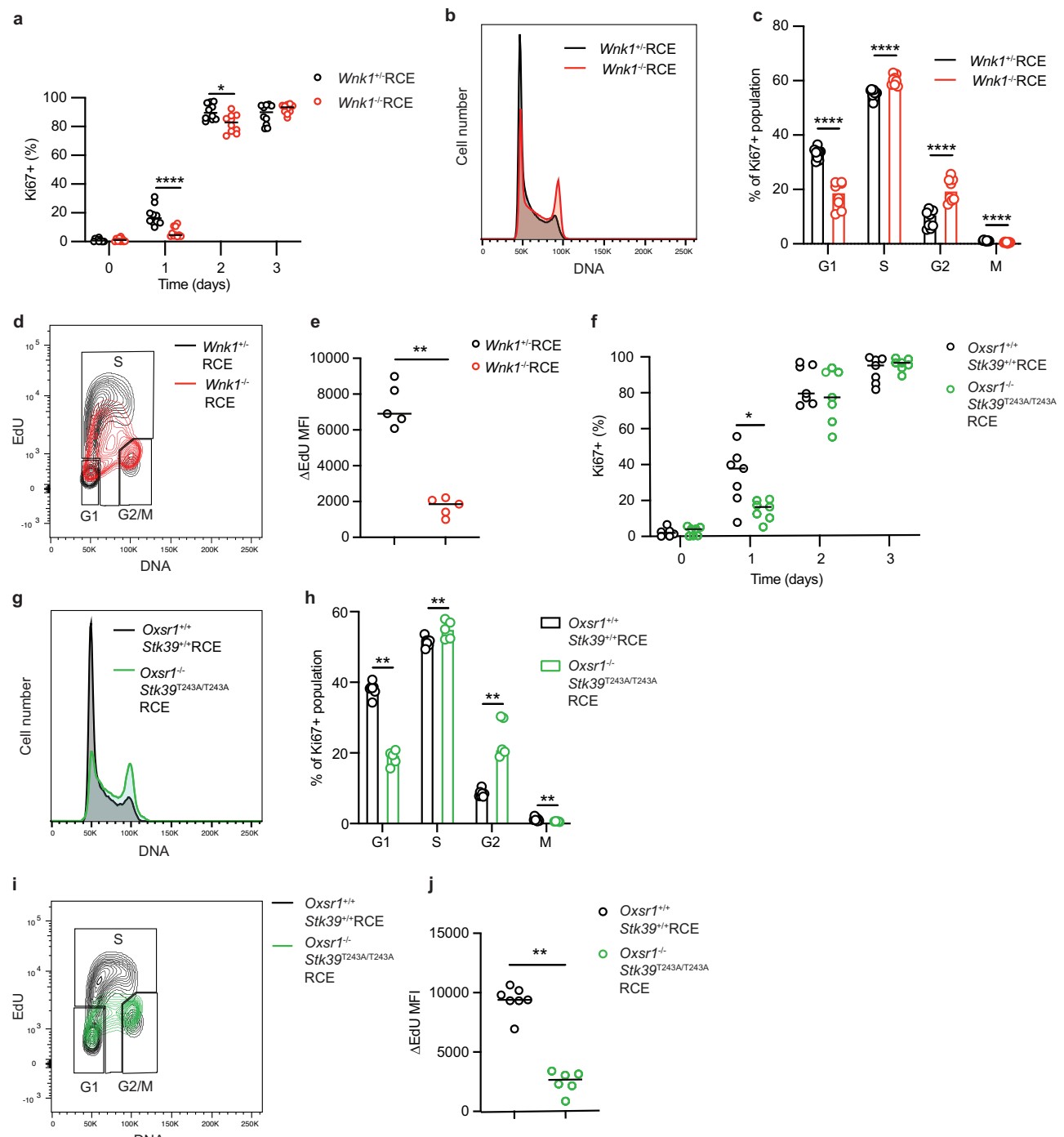

**Fig. 3 | Loss of WNK1, or OXSR1 and STK39 impairs cell cycle progression of CD4⁺ T cells. a–j** Naive CD4⁺ T cells of the indicated genotypes, or treated with indicated inhibitors, were activated on plate-bound anti-CD3ε and anti-CD28 antibodies for 3 days. Graphs show percentage of Ki67⁺ cells (after adjusting for population expansion) (**a, f**), and DNA content of Ki67⁺ cells (**b, g**) which was used to estimate the median ($\pm$ 95% CI) percentage of cells in different cell cycle phases at day 3 (**c, h**). Cells were pulsed with EdU for 10 min and incorporation of EdU (**d, i**) was used to calculate the rate of EdU incorporation (ΔEdU) by subtracting the average EdU MFI in G1 and G2/M phase cells from the average EdU MFI in S phase

(**e, j**) using the gates shown. Sample numbers: **a**, $n = 10$; **c**, *Wnk1*⁺/⁻RCE: $n = 9$, *Wnk1*⁻/⁻RCE: $n = 8$; e, $n = 5$; **f**, $n = 7$ except *Oxsr1*⁺/⁺*Stk39*⁺/⁺RCE at day 0, $n = 6$; **h**, *Oxsr1*⁺/⁺*Stk39*⁺/⁺RCE: $n = 7$, *Oxsr1*⁻/⁻*Stk39*^T243A/T243A^RCE: $n = 5$; **j**, *Oxsr1*⁺/⁺*Stk39*⁺/⁺RCE: $n = 7$, *Oxsr1*⁻/⁻*Stk39*^T243A/T243A^RCE: $n = 6$. Each point represents a different mouse; horizontal lines or columns show median; data pooled from 2 independent experiments (**a, c, f, h, j**) or from 1 experiment representative of 2 (**e**) experiments. Statistical analysis with the 2-sided Mann-Whitney U test. * 0.01 <$p$< 0.05; ** 0.001 <$p$< 0.01, **** $p$ < 0.0001. Source data are provided as a Source Data file.

signals in our system. Flow cytometric analysis of CD4⁺ T cells stimulated with anti-CD3ε and anti-CD28 antibodies in solution showed that WNK1 inhibition had no effect on the phosphorylation of the CD3ζ component of the TCR on Y142 (p-CD3ζ), one of the earliest measurable biochemical changes following TCR stimulation (Fig. 4a,

Supplementary Fig. 5a). As expected, this TCR/CD28-induced increase in p-CD3ζ was blocked by treatment with a SRC kinase inhibitor (SRCi). In contrast, WNKi treatment caused a significant reduction in the phosphorylation of the ZAP70 kinase on Y319 (p-ZAP70) and the ERK kinases on T202 and Y204 (p-ERK), both of which were also blocked by

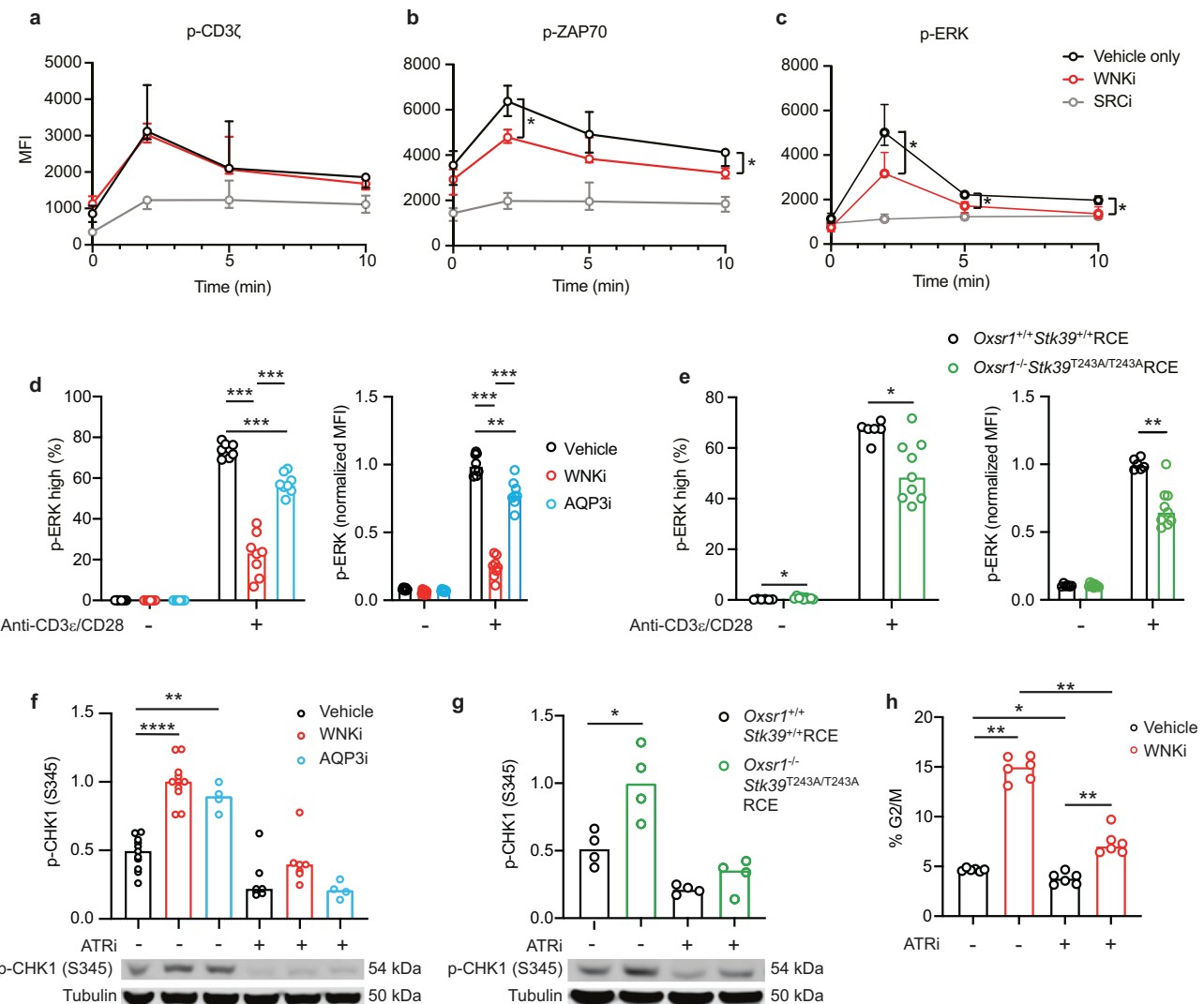

**Fig. 4 | The WNK1 pathway regulates early TCR signaling and the ATR-mediated G2/M checkpoint. a–c** Naïve WT CD4[+] T cells were activated using soluble anti-CD3ε and anti-CD28 antibodies in the presence of indicated inhibitors and analyzed by flow cytometry. Graphs show median ( ± 95% CI) levels of p-CD3ζ (Y142) (**a**), p-ZAP70 (Y319) (**b**), and p-ERK1/2 (T202/Y204) (**c**) (*n* = 4); example histograms of flow data shown in Supplementary Fig. 5a-c. **d, e** CD4[+] T cells of the indicated genotypes or in the presence of the indicated inhibitors were cultured for 1 h in the presence or absence of plate-bound anti-CD3ε and anti-CD28 antibodies. Graphs show the percentage of cells that were p-ERK high and p-ERK mean fluorescence intensity normalized to stimulated controls cells (set to 1) as determined by flow cytometry, gating as in Supplementary Fig. 5d, e (**d**, *n* = 8; **e**, *Oxsr1*[+/+]*Stk39*[+/+]RCE: *n* = 6, *Oxsr1*[+/+]*Stk39*[T243A/T243A]RCE: *n* = 9). **f–h** CD4[+] T cells were cultured for 3 d on plate-bound anti-CD3ε and anti-CD28 antibodies, with ATRi or vehicle only added

to cultures for the final 36 h. Graph of the abundance of p-CHK1 (S345) determined using immunoblots (example immunoblots below graph, sizes in kDa) (**f**, columns 1, 2: *n* = 11, columns 3, 6: *n* = 4, columns 4, 5: *n* = 7; **g**, *n* = 4). p-CHK1 was normalized to tubulin and to the mean signal in WNKi-treated cells (**f**) or *Oxsr1*[-/-]*Stk39*[T243A/T243A]RCE cells (**g**) which was set to 1. Graph showing percentage of CD4[+] cells in G2/M phase (EdU[-], 4n), gating as in Supplementary Fig. 4e (**h**, *n* = 6). WNKi, WNK inhibitor (WNK463); SRCi, SRC inhibitor (Dasatinib); AQP3i, AQP3 inhibitor (DFP00173); ATRi, ATR inhibitor (AZD6738). Each point represents a different mouse; columns show median. Data pooled from 2 (**a-d, g, h**) or 3 (**e**) independent experiments or from 1 experiment representative of 5 (**f**). Statistical analysis carried out using the 2-sided Mann-Whitney U test. * 0.01 < *p* < 0.05; ** 0.001 < *p* < 0.01, *** 0.0001 < *p* < 0.001, **** *p* < 0.0001. Source data are provided as a Source Data file.

SRCi (Fig. 4b, c, Supplementary Fig. 5b, c). Moreover, p-ERK was reduced in WNKi-treated CD4[+] T cells activated by plate-bound anti-CD3ε and anti-CD28 antibodies for 1 h, conditions that result in biphasic activation of ERK and T cell proliferation (Fig. 4d, Supplementary Fig. 5d)[17]. A similar decrease in TCR/CD28-stimulated p-ERK was also seen in *Oxsr1*[-/-]*Stk39*[T243A/T243A]RCE T cells (Fig. 4e, Supplementary Fig. 5e), implying that WNK1-OXSR1-STK39 signaling is required for TCR/CD28-induced p-ERK.

### Defects in the WNK1 pathway result in an ATR-dependent cell cycle arrest in G2

Accumulation of cells in G2 could be caused by the triggering of a cell cycle checkpoint, for example through activation of the ATM or ATR

kinases in response to DNA damage. Immunoblot analysis of phosphorylated RPA32 (S4/8, p-RPA32) and phosphorylated H2AX (S139, γH2AX), markers of single-stranded and double-stranded DNA damage respectively, showed no significant elevation of either in WNK1-inhibited T cells stimulated through the TCR and CD28 (Supplementary Fig. 4c, d). However, as expected, both were robustly increased by treatment with hydroxyurea, which causes replication fork stalling and subsequent DNA damage. Since γH2AX is a marker of ATM activity, these results imply that inhibition of WNK1 does not result in ATM activation. To measure ATR activity, we immunoblotted cell extracts for phosphorylated CHK1 (S345, p-CHK1), a product of ATR activity. Unexpectedly, we detected elevated p-CHK1 in TCR/CD28-stimulated WNKi-treated T cells compared to control

T cells, which was reduced in cells treated with AZD6738, an ATR inhibitor (ATRi), indicating that ATR activity was increased in the absence of WNK1 activity (Fig. 4f). Extending this analysis, we found a similar increase in ATR activity in TCR/CD28-stimulated *Oxsr1*[-/-]*Stk39*[T243A/T243A]RCE T cells (Fig. 4g). To evaluate if increased ATR activity in WNK1-inhibited T cells was causing cell cycle arrest, we activated WNK1-inhibited T cells and treated them with ATRi. We found that the inhibitor significantly reduced the fraction of cells in G2, implying that the block in G2 is predominantly due to ATR activation (Fig. 4h, Supplementary Fig. 4e). Thus, compromised TCR/CD28 signaling via WNK1, OXSR1 and STK39 leads to ATR activation and cell cycle arrest.

### Hypertonic activation of WNK1 is not sufficient to cause T cell activation

The previous experiments had shown that WNK1 activity was required for TCR/CD28-induced activation of CD4+ T cells. To determine if WNK1 activity alone was sufficient to activate T cells, we evaluated T cell activation following treatment of the cells with hypertonic medium which has been shown to cause WNK1 activation in other cell types[9]. As expected, hypertonic medium induced WNK1 activation as seen by increased phosphorylation of OXSR1 on Ser325 (Supplementary Fig. 6a, b). However, hypertonic medium on its own was not able to induce an increase in p-ERK, and did not result in any cells inducing expression of Ki67 or going into cell division (Supplementary Fig. 6c, d). Indeed, a higher osmolarity of 400 mOsm/L partially suppressed anti-CD3ε/anti-CD28-induced T cell activation. These results show that WNK1 activation alone is not sufficient to cause T cell activation.

### Na+ and Cl- ions are important for TCR/CD28-induced ERK activation

In kidney epithelial cells, signaling through WNK1, OXSR1 and STK39 leads to phosphorylation of SLC12A-family ion co-transporters and net influx of Na+, K+ and Cl- ions[6]. Thus, we hypothesized that TCR signaling via the WNK1-OXSR1-STK39 pathway leads to SLC12A-dependent ion influx which is required for T cell activation. To test whether WNK1 regulates ion transport, we measured chloride-dependent thallium influx as a surrogate for chloride-dependent K+ influx, a characteristic of ion co-transporters such as SLC12A2. This showed that TCR/CD28 stimulation results in increased K+ influx which is blocked by treatment with WNKi (Fig. 5a). To test whether ions are important for TCR-ERK signaling, we reduced the extracellular concentration of Na+ and Cl- ions while maintaining isotonicity with L-glucose, an inert osmolyte. This impaired TCR-CD28-stimulated p-ERK in CD4+ T cells (Fig. 5b). Furthermore, replacement of Cl- ions in the medium with gluconate, also strongly reduced TCR/CD28-induced p-ERK (Fig. 5c). This reduced p-ERK activation is unlikely to have been caused by changes in membrane potential, since this showed little or no change in media with reduced Na+ and Cl- used in Fig. 5b and c (Supplementary Fig. 7a, b). Taken together, these results show that TCR signaling causes WNK1-dependent ion influx and imply that Na+ and Cl- ions are likely important for TCR/CD28 signaling.

To examine the role of the SLC12A-family ion co-transporters in this process, we initially investigated their expression in T cells. RNA sequencing (RNAseq) showed that of the 9 family members, 5 were expressed in mouse CD4+ T cells: *Slc12a2*, *Slc12a4*, *Slc12a6*, *Slc12a7* and *Slc12a9* (Fig. 5d). Furthermore, flow cytometric analysis showed that TCR/CD28 stimulation resulted in increased phosphorylation of

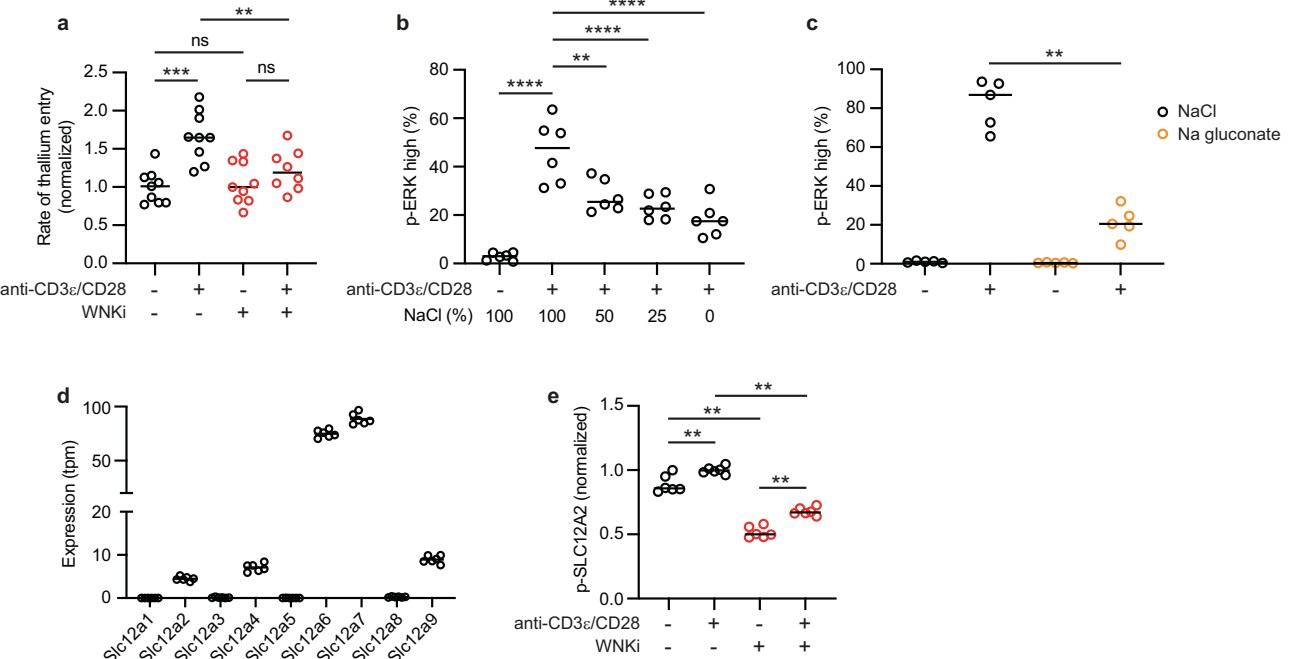

**Fig. 5 | Na and Cl are required for TCR-induced activation. a** Rate of thallium entry as a measure of chloride-dependent K+ flux in CD4+ T cells unstimulated or stimulated with beads coated with anti-CD3ε and anti-CD28 antibodies, with or without WNKi. Rate was normalized to unstimulated cells in the absence of inhibitor (set to 1) (n = 9, except column 4, n = 8). **b, c** CD4+ T cells were cultured for 1 h in the presence or absence of plate-bound anti-CD3ε and anti-CD28 antibodies in medium in which different proportions of NaCl were replaced with L-glucose (**b**) or where the NaCl was replaced with Na gluconate (**c**), in both cases maintaining isotonicity. Note that the medium contains Na+ and Cl- from other constituents other than NaCl. Graphs show percentage p-ERK high cells (b, n = 6; c, n = 5).

**d** mRNA levels of *Scl12a* genes in *Wnk1*[+/-] CD4+ T cells determined by RNAseq; tpm, transcripts per million reads (n = 6). **e** Levels of p-SLC12A2 (T203, T207, T212) in CD4+ T cells cultured for 1 h in the presence or absence of plate-bound anti-CD3ε and anti-CD28 antibodies, with or without WNKi, measured by flow cytometry and normalized to the mean signal in stimulated cells without inhibitor (n = 6). WNKi, WNK inhibitor (WNK463). Each point represents a different mouse; lines show median. Data is pooled from 3 experiments (**a**) or 2 experiments (**b, c, e**) or is from 1 experiment (**d**). ns, not significant. 2-sided Mann-Whitney U test; ** 0.001 < p < 0.01, **** p < 0.0001. Source data are provided as a Source Data file.

SLC12A2 on Thr203, Thr207 and Thr212 (p-SLC12A2), sites that are phosphorylated by OXSR1 and STK39[8] (Fig. 5e, Supplementary Fig. 8a). Treatment of CD4+ T cells with WNKi resulted in decreased phosphorylation of SLC12A2, in support of the hypothesis that a WNK1-OXSR1-STK39 pathway leads to SLC12A2 phosphorylation in T cells. To investigate the role of SLC12A2 in T cell activation we made use of bumetanide, an SLC12A2 inhibitor (SLC12A2i) which we had previously shown inhibits mouse CD4+ T cell migration[10,18], and also examined SLC12A2-deficient T cells. We found that treatment with SLC12A2i or loss of SLC12A2 did not affect TCR-CD28-induced p-ERK or the percentage of CD4+ T cells that divided at least once (Supplementary Fig. 8b-e).

Since SLC12A2 is only one of 5 SLC12A-family members that is expressed in T cells, several of which are known substrates of the WNK1-OXSR1-STK39 pathway[4], other SLC12A members may be more important than SLC12A2 in transducing WNK1 pathway signals that regulate T cell activation and proliferation. Despite their structural similarity, the family members have different, opposing functions. While WNK1-OXSR1-STK39-induced phosphorylation of SLC12A2 (NKCC1) leads to its activation and hence influx of $Na^+$, $K^+$ and $Cl^-$ ions, phosphorylation of SLC12A6 (KCC3) or SLC12A7 (KCC4) inhibits these transporters, blocking efflux of $K^+$ and $Cl^-$ ions, with the net result being an increase in the intracellular concentrations of $Na^+$, $K^+$ and $Cl^-$ ions[5]. Bearing in mind this complication, we examined the effect of DIOA, an inhibitor of SLC12A6 and SLC12A7 (SLC12A6/7i) on T cell activation. Once again, we found that treatment of T cells with SLC12A6/7i did not affect TCR/CD28-induced p-ERK or cell division, possibly because the inhibitor is simply reinforcing their physiological inhibition by WNK1 activity downstream of the TCR (Supplementary Fig. 8b-e). Taken together, these results indicate that TCR/CD28 signaling leads to WNK1-dependent ion influx and that $Na^+$ and $Cl^-$ ions are important for TCR/CD28-induced T cell activation, but we were unable to establish if this was due to a role for SLC12A-mediated ion influx.

## AQP3 is required for TCR-induced activation

One of the effects of ion influx into the cell is that the resulting osmotic gradient causes passive water influx. Indeed, WNK1 plays an important role in the osmoregulation of cells and in modulating cell volume by regulating ion influx and consequent osmotically-driven water influx[9]. Thus, we hypothesized that TCR signaling through WNK1 would result in water influx and that this may be required for T cell activation. In support of this, stimulation of WT naive CD4+ T cells with anti-CD3ε and anti-CD28 resulted in a cell volume increase as measured using a Casy® cell counter, whereas WNKi-treated cells were smaller and did not increase in volume to the same extent (Fig. 6a). We extended this analysis by directly measuring the volume of the cells by super-resolution microscopy. This showed that TCR/CD28 stimulation resulted in an ~11% increase in cell volume, whereas WNKi treatment reduced basal cell volume by ~22% and eliminated the TCR-induced increase in volume (Fig. 6b, c). In contrast, treatment of CD4+ T cells with SLC12A2i did not affect basal or TCR/CD28-induced volume, in agreement with the lack of effect of this inhibitor on TCR-induced p-ERK and proliferation (Supplementary Fig. 8f). Thus, WNK1 is required for cell volume increase in response to TCR and CD28 stimulation, consistent with our hypothesis that TCR signaling via WNK1 results in water influx.

Water entry into mammalian cells is mediated in part by aquaporins, channels that allow water to passively flow into or out of cells in response to an osmotic gradient[19]. RNA sequencing analysis showed that of the 11 functional aquaporin genes in mice, CD4+ T cells express detectable levels of only three family members: *Aqp3*, *Aqp9* and *Aqp11* (Fig. 6d). Since of these three, *Aqp3* is expressed at the highest level, we decided to use DFP00173, an AQP3 inhibitor (AQP3i)[20], to test the requirement for water entry in T cell activation. As expected, AQP3i reduced the water permeability of CD4+ T cells (Fig. 6e). Notably,

treatment of naive CD4+ T cells with AQP3i significantly reduced TCR-CD28-induced proliferation (Fig. 6f, g). Furthermore, AQP3i treatment also resulted in reduced TCR/CD28-induced p-ERK induction following 1 h of stimulation, as well as increased p-CHK1 after 3 d of stimulation (Fig. 4d, f), similar to the phenotype caused by inhibition or mutation of WNK1, OXSR1 and STK39.

To extend this study we made use of mice bearing the *Aqp3*[tm2a(EUCOMM)Wtsi] (*Aqp3*[tm2a]) allele which has lacZ and neo genes inserted into intron 1 of the *Aqp3* gene along with a splice acceptor and poly adenylation sites (Supplementary Fig. 3d)[21]. Typically, alleles with this structure show loss of function because of the integrated splice acceptor and poly adenylation sites. In line with this, *Aqp3*[tm2a/tm2a] mice homozygous for this allele had a 95-98% reduction of *Aqp3* mRNA with no significant change in the expression of *Aqp9* or *Aqp11* (Supplementary Fig. 3e). *Aqp3*[tm2a/tm2a] CD4+ T cells from both lymph nodes and spleen showed significantly reduced TCR/CD28-induced p-ERK compared to wild-type control cells, in agreement with the results obtained using AQP3i, albeit the effect of the genetic loss of *Aqp3* was not as large as that of the inhibitor (Fig. 6h, i, Supplementary Fig. 5f, g). Furthermore, *Aqp3*[tm2a/tm2a] CD4+ T cells had reduced TCR/CD28-induced upregulation of Ki67 at 24 h, although cell division at 48 h was unaffected (Fig. 6j, k). The stronger effect of AQP3i compared to the genetic mutant may be due to residual AQP3 expression in *Aqp3*[tm2a/tm2a] CD4+ T cells, to off-target effects of AQP3i on other AQPs or other proteins, or to adaptation of *Aqp3*[tm2a/tm2a] cells to the chronic lack of AQP3 activity during CD4+ T cell development. Nonetheless, the results support a role for AQP3 in TCR/CD28-induced activation of ERK and entry into G1 as measured by Ki67 upregulation.

In addition to water, AQP3 is also a channel for $H_2O_2$, a molecule which can modulate intracellular signaling pathways[22]. To test whether the requirement for AQP3 in T cell activation was due to its ability to facilitate diffusion of $H_2O_2$ across membranes, we investigated if $H_2O_2$ is required for TCR/CD28-induced p-ERK. Initially we determined that 20 u/ml of catalase was sufficient to eliminate exogenously added 1 mM $H_2O_2$ (Supplementary Fig. 9a). Subsequently, we showed that 20 u/ml catalase did not affect the TCR/CD28-induced p-ERK after 1 hour of stimulation, and the same was seen at 2 and 200 u/ml (Supplementary Fig. 9b). Thus, we find no evidence that $H_2O_2$ is required for TCR-induced signaling to ERK activation. AQP3 can also transport glycerol[19]. However, this function is unlikely to account for the defect in TCR/CD28-induced p-ERK and proliferation in AQP3i-treated T cells or in impaired induction of p-ERK and Ki67 in AQP3-deficient T cells, since there is no glycerol in the medium. Taken together, these results support our hypothesis that TCR signaling through WNK1, OXSR1 and STK39 results in water entry, potentially through AQP3, which is required for early TCR signaling, entry into G1 and suppression of the ATR-mediated G2/M checkpoint.

## WNK1-dependent water influx is required for TCR signaling and G1 entry

To directly test whether TCR/CD28 signaling through WNK1 leads to water entry that is required for T cell activation, we stimulated WNK1-deficient CD4+ T cells in hypotonic medium, to force water entry by osmosis. Remarkably, we found that hypotonic medium restored much of the reduced TCR/CD28-stimulated p-ERK in WNK1[D368A]-expressing T cells, and all of the p-ERK in WNKi-treated T cells as measured 1 h after stimulation (Fig. 7a, b, Supplementary Fig. 10a, b). Furthermore, hypotonic medium also rescued the defect in TCR/CD28-induced Ki67 upregulation at 2 d (Fig. 7c, Supplementary Fig. 10c). The effect of hypotonic medium could be due to increased water entry into the cell but could also be caused by the reduced concentration of $Na^+$ and $Cl^-$ ions in the medium. To distinguish these possibilities, we stimulated the T cells in isotonic medium that had the same reduced concentrations of $Na^+$ and $Cl^-$ but the osmolarity was made isotonic with L-glucose, an inert osmolyte. Importantly, this

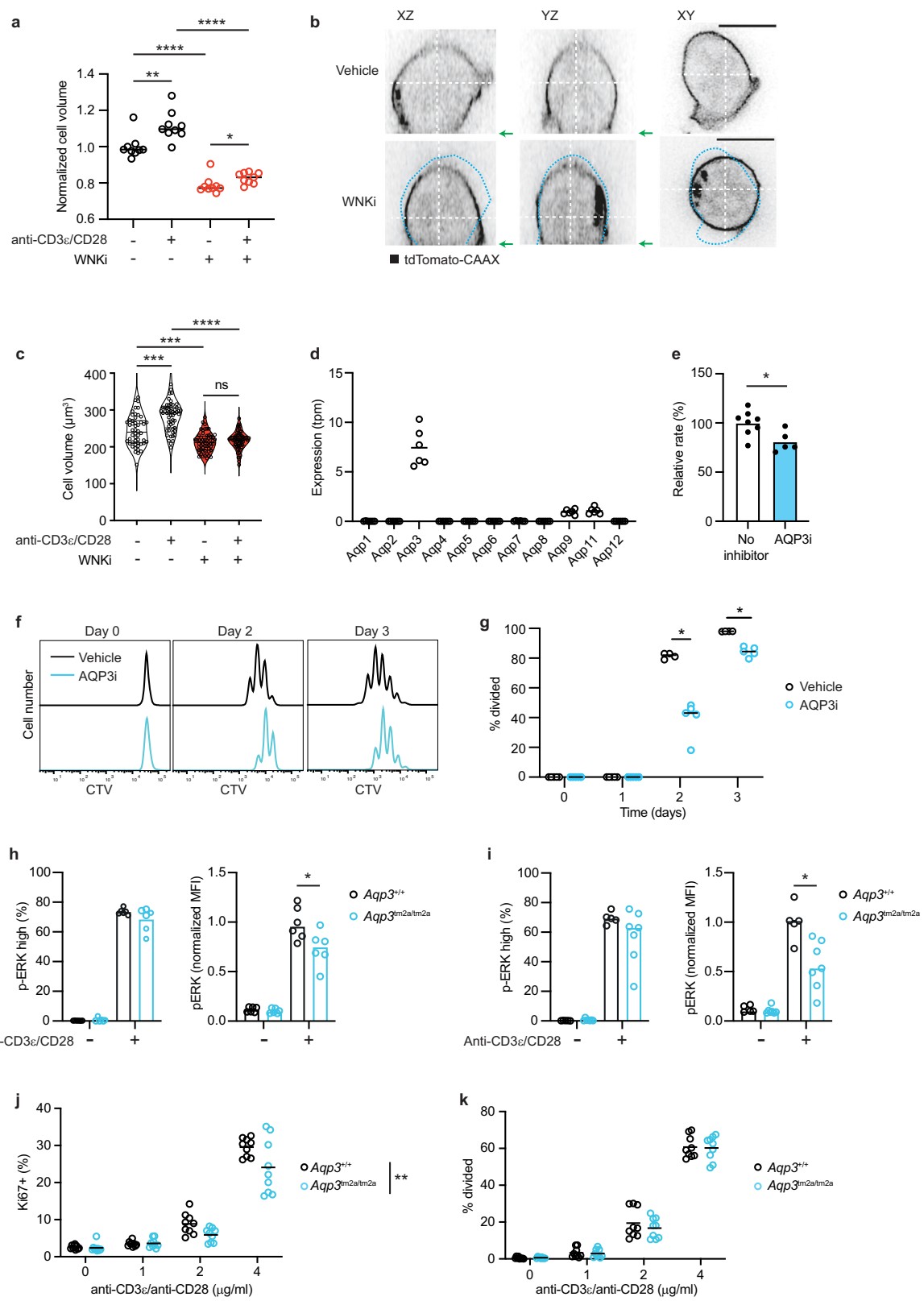

isotonic medium with reduced Na⁺ and Cl⁻ concentrations was unable to rescue the TCR/CD28-stimulated p-ERK at 1 h or the induction of Ki67 at 2 d in WNK1$^{D368A}$-expressing or WNKi-inhibited CD4⁺ T cells (Fig. 7a–c, Supplementary Fig. 10a-c). Thus, the rescue seen with hypotonic medium was due to water entry, not reduced ion concentrations. Taken together, these results demonstrate that TCR/CD28 co-stimulation of CD4⁺ T cells leads to water entry mediated by the

WNK1-OXSR1-STK39-AQP3 signaling pathway, and this is required for TCR/CD28-induced ERK activation and entry into G1 (Fig. 7d).

## Discussion

We show that WNK1 signaling through OXSR1, STK39 and potentially AQP3 plays a central role in CD4⁺ T cell proliferation in response to antigen, and thus the generation of a class-switched antibody

**Fig. 6 | WNK1-dependent volume changes and AQP3-dependent TCR-induced proliferation. a** Volume of naïve WT CD4+ T cells activated for 1 h in the presence of vehicle only or WNKi on plate-bound anti-CD3ε and anti-CD28 antibodies where indicated, determined using a Casy® cell counter and normalized to the mean volume of cells in the absence of stimulation or inhibitor. Each dot represents the modal volume of ≥200 cells (n = 9). **b** Orthogonal sections of 3D images of mouse naïve CD4+ T cells constitutively expressing tdTomato-CAAX, stimulated for 1 h on plate-bound anti-CD3ε and anti-CD28 antibodies with WNKi or vehicle only. Grey scale represents the tdTomato fluorescence intensity. White dashed lines show positions of the orthogonal slices which were taken at the widest and highest point of the cell. Blue dashed line represents the outline of the vehicle control cell superimposed on the WNKi-treated cell. Green arrows show the location of the cell/slide interface in the XZ and YZ sections. Scale bars, 5 μm. **c** Cell volume measured by imaging of CD4+ T cells stimulated for 1 h with plate-bound anti-CD3ε and anti-CD28 antibodies as indicated, with or without WNKi, representative images shown in **b**. Each dot represents 1 cell, violin plots show median and 25% and 75% quartiles (columns 1-4, n = 50,55,56,60). **d** mRNA levels of *Aqp* genes in *Wnk1*+/- CD4+ T cells determined by RNAseq; tpm, transcripts per million reads (n = 6). **e** Water permeability of CD4+ T cells treated with or without AQP3i determined by measuring volume changes in response to an inwardly-directed 150 mM osmotic gradient.

Graph shows relative rate constant normalized to the mean of cells with no inhibitor. Column shows mean (no inhibitor, n = 8; AQP3i, n = 5). **f, g** Naïve CD4+ CTV-labelled T cells treated with AQP3i were activated on plate-bound anti-CD3ε and anti-CD28 antibodies for indicated times. CTV dilution (**f**) was used to determine the percentage of cells that had divided (n = 5, except Vehicle days 2-3, n = 4) (**g**). **h, i** CD4+ T cells of the indicated genotypes from lymph nodes (**h**) or spleen (**i**) were cultured for 1 h in the presence or absence of plate-bound anti-CD3ε and anti-CD28 antibodies. Graphs show the percentage of cells that were p-ERK high and p-ERK mean fluorescence intensity normalized to stimulated controls cells (set to 1) as determined by flow cytometry, gating as in Supplementary Fig. 5f, g (**h**, n = 6; i, *Aqp3*+/+: n = 5, *Aqp3*tm2a/tm2a: n = 7). **j, k** Lymph node CD4+ T cells of the indicated genotypes were activated on plate-bound anti-CD3ε and anti-CD28 antibodies at the indicated doses. Graphs show the percentage of Ki67+ cells at 24 h (**j**) and the percentage of cells that had divided at 48 h (**k**) (n = 9). Dots represent biological replicates. Data are pooled from 2 independent experiments (**a, c, e, h-k**), from 1 experiment representative of 2 (**g**) or from a single experiment (**d**). WNKi, WNK inhibitor (WNK463); AQP3i, AQP3 inhibitor (DFP00173). Statistical analysis with the 2-sided Mann-Whitney test (**a, e, g-i**), Kruskal-Wallis test with correction for multiple comparisons (**c**) or 2-way ANOVA (**j**). * $0.01 < p < 0.05$; ** $0.001 < p < 0.01$. *** $0.0001 < p < 0.001$. **** $p < 0.0001$. Source data are provided as a Source Data file.

response. WNK1 pathway activity is required for entry of naive CD4+ T cells into G1 and progression through S and G2/M. More specifically, WNK1-dependent water influx, a previously unexplored component of the TCR signaling cascade, is required for TCR-induced ERK signaling and entry into G1. These two WNK1-dependent, water-sensitive processes are probably linked as progression from G0 to G1 is strongly dependent on ERK activation[23].

Abrogation of WNK1 activity causes larger defects in TCR-induced p-ERK activation and proliferation than mutations in *Oxsr1*, *Stk39*, and *Aqp3* or inhibition of AQP3, suggesting that WNK1 functions in part through other pathways to regulate these processes. The difference in magnitude may also be due to redundancy with signaling proteins functioning in parallel with OXSR1, STK39, and AQP3. For example, water flux through AQP9 and AQP11 may compensate for the loss of AQP3 activity. Furthermore, some of the effects of chronic loss of WNK1 may be due to altered expression of osmoregulatory proteins, such as AQP9 (Supplementary Fig. 1f). Nonetheless, we note that hypotonic medium rescues the defective TCR-induced p-ERK activation at 1 hour of stimulation and Ki67 induction after 2 days in WNKi-treated T cells. This strongly supports our proposal that TCR signaling through WNK1 results in water entry that is required for early TCR signaling and the transition from G0 to G1.

It is interesting to note that Cl- efflux through volume-regulated chloride channels in hypotonic NaCl-low conditions could also contribute to the rescue of TCR/CD28-induced ERK signaling in WNK1-deficient cells, a possibility that would need further investigation[24].

We hypothesize that TCR/CD28 signaling via WNK1 and OXSR1 and STK39 leads to SLC12A-dependent ion influx, which in turn drives water entry by osmosis through AQP3. In agreement with this, we showed that TCR/CD28 stimulation leads to an increase in chloride-dependent K+ flux which was blocked by WNKi. Furthermore, we showed that TCR/CD28-induced ERK activation requires Na+ and Cl- ions and that WNK1 is required for SLC12A2 phosphorylation during CD4+ T cell activation, but we were not able to demonstrate a functional requirement for SLC12A2, SLC12A6 or SLC12A7 in this process using inhibition and/or genetic deletion. This could be because we did not mutate or inhibit the key SLC12A-family members of the 5 expressed in CD4+ T cells, or because other WNK1-regulated ion channels are more important. Beyond the SLC12A-family, WNK kinases have been reported to regulate several other channels including ENaC, CLC3 and TRP6[25]. It is also possible that the WNK1-regulated ion influx has other roles in TCR signaling beyond causing water entry by osmosis. Further work will be required to address these possibilities.

In the absence of WNK1 activity, T cells are smaller, and the TCR-induced volume increase is reduced compared to control cells. The volume decrease in WNKi-treated cells, even in the absence of TCR stimulation, demonstrates a homeostatic role for WNK1 in controlling T cell volume. Indeed, WNK kinases are important for the regulatory volume increase that follows cell shrinkage induced by hypertonic conditions[4,9]. In the hypotonic rescue experiments, we propose that the resulting water entry would compensate for the defective regulatory volume increase in WNKi-inhibited T cells, as well as the defect in the additional WNK1-dependent TCR-induced volume increase.

Inhibition of AQP3 resulted in decreased TCR/CD28-induced p-ERK and proliferation. However, the genetic loss of *Aqp3* gave a milder phenotype with a smaller decrease in p-ERK and Ki67 upregulation, and no effect on cell division. This difference could be due to several factors. There may be residual expression of AQP3 in the genetic mutant facilitating flux through the TCR-ERK signaling pathway and T cell proliferation. Alternatively, the *Aqp3*tm2a/tm2a cells may have adapted to the chronic loss of AQP3 by altering expression of other proteins, or the inhibitor may act on the other AQP proteins or have other off-target effects. Further work will be needed to distinguish these possibilities. Nonetheless, our results show that water entry is required for TCR/CD28 induced p-ERK and Ki67 upregulation and that the water is likely to enter in part through AQP3.

Water entry alters several biophysical properties of the cell which could affect TCR signaling. Water influx increases membrane tension which has been shown to regulate signaling pathways[26–30]. Water entry also reduces molecular crowding, thereby increasing rates of intracellular signaling pathways[31], and decreases the viscosity of the cytoplasm, resulting in more rapid assembly of macromolecular structures such as microtubules[32]. Any or all of these changes could facilitate the assembly of the macromolecular TCR signaling complexes at the immune synapse and subsequent signal transduction[33]. Similarly, water entry could be required to facilitate the function of the macromolecular replisome complex, thereby accounting for the decreased rate of DNA replication in WNK1-deficient T cells. Furthermore, entry of water into T cells is presumably required to compensate for the rapid synthesis of macromolecules that accompanies T cell activation and entry into G1[34].

We also show that in the absence of WNK1-OXSR1-STK39-AQP3 signaling, ATR is activated and induces cell cycle arrest in G2. After 24 h, T cell division is no longer dependent on TCR signaling[35]. Thus, the requirement for water influx for T cells to progress through an ATR-dependent G2 checkpoint 72 h after the start of T cell activation is

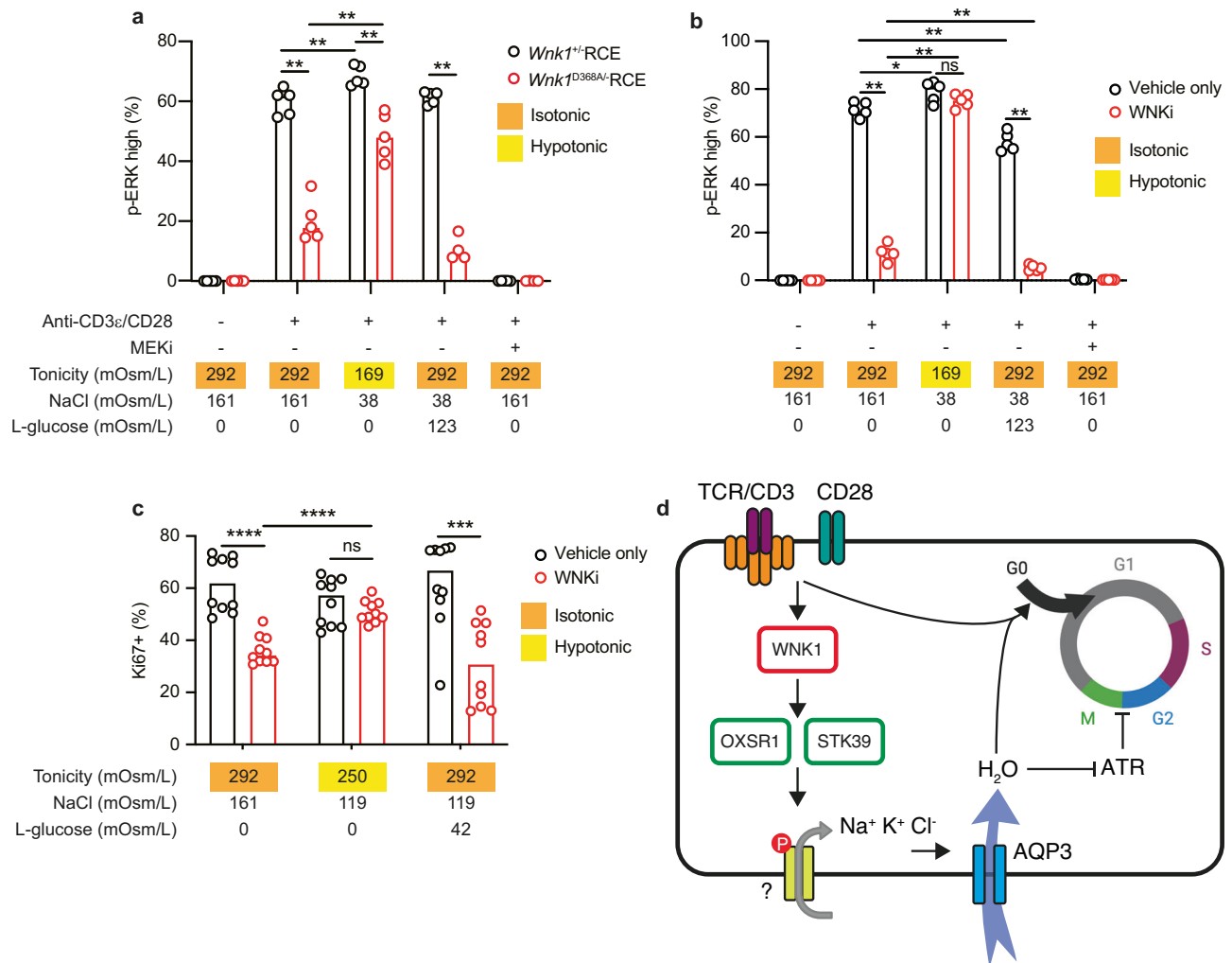

**Fig. 7 | WNK1-dependent water influx is required for TCR signaling and G1 entry. a, b** CD4+ T cells of the indicated genotypes or in the presence of the indicated inhibitors were cultured for 1 h in the presence or absence of plate-bound anti-CD3ε and anti-CD28 antibodies. In some cultures, tonicities were altered by modulating concentrations of NaCl and L-glucose. Graphs show the percentage of cells that were p-ERK high as determined by flow cytometry, gating as in Supplementary Fig. 10a, b (*n* = 5). **c** CD4+ T cells were cultured on plate-bound anti-CD3ε and anti-CD28 antibodies for 2 days at the indicated tonicities. Graph shows percentage of Ki67+ cells (after adjusting for population expansion) measured by flow cytometry, gating as in Supplementary Fig. 10c (*n* = 10). **d** Proposed model of how TCR/CD28 signals via WNK1, OXSR1 and STK39, via unknown ion transporters

(possibly SLC12A-family), lead to ion influx and subsequent water entry through AQP3 by osmosis, which is required for TCR/CD28 signaling and T cell proliferation. Water entry is required for early TCR signaling and for the transition from G0 to G1. In addition, the WNK1 pathway inhibits ATR and thereby allows cells to transit the G2/M checkpoint. WNKi, WNK inhibitor (WNK463). Each point represents a different mouse; columns show median. Data pooled from 2 (**c**) or from 1 experiment representative of >2 (**a, b**) experiments. Statistical analysis carried out using the 2-sided Mann-Whitney U test. ns, not significant; * 0.01 <*p* < 0.05; ** 0.001 <*p* < 0.01; *** 0.0001 <*p* < 0.001; **** *p* < 0.0001. Image of cell cycle created in Biorender. Tybulewicz, V. (2024) https://BioRender.com/k55d532. Source data are provided as a Source Data file.

likely to be independent of TCR signaling. This TCR-independent function of the WNK1 pathway may reflect a homeostatic role in the regulation of cellular water content and volume or a specific role at the G2/M transition.

We found that inhibition of WNK1 or AQP3 resulted, unusually, in ATR activation independently of DNA damage. ATR is activated by mechanical stress at the nuclear envelope caused by a change in cell volume[36]. A similar mechanism may account for ATR activation in T cells in which the WNK1 pathway is inhibited, since this results in cell shrinkage. Interestingly, other studies have shown that compromised osmoregulation causes G2 arrest in multiple cell types, including T cells[37,38]. Thus, we have uncovered an ATR-dependent mechanism for a potentially universal osmotically-sensitive cell cycle checkpoint at G2/M, a possibility that warrants further study.

WNK1 may additionally regulate T cell proliferation by other mechanisms. In HeLa cells, WNK1 was shown to regulate the

formation of mitotic spindles[39]. Furthermore, WNK1 is a negative regulator of autophagy in U2OS and HeLa cells, a cellular process that inhibits proliferation[40–42]. Hypotonic medium leads to the proliferation of vascular smooth muscle cells via the activation of WNK1[24]. Finally, WNK1 regulates protein trafficking, so it may transduce signals by altering surface expression of AQP3 or other proteins required for proliferation[43–45]. These are interesting areas for future investigation.

Several observations suggest that WNK1-dependent water influx may be a common feature of mitogenic pathways in many cell types, both within the immune system and beyond. First, WNK1 is broadly expressed[46]. Second, water influx occurs during mammalian cell division[47–49], and, thirdly, aquaporins are required for cell proliferation in several cell types[50–53]. Furthermore, we have recently found that WNK1 is activated in B cells by stimulation through the B cell antigen receptor and CD40, and that WNK1, OXSR1 and STK39 are

required for proliferation induced by signals from both receptors[54]. Taken together with our results, we propose that WNK1 pathway-driven water influx may be essential for the proliferation of multiple cell types.

More broadly, in a parallel study we have shown that chemokine receptor signaling in CD4+ T cells activates the WNK1 pathway, leading to ion and water influx and that water entry is required for T cell migration[18]. This shows that activation of WNK1 and downstream ion and water influx is an important feature of signaling from several receptors and that it plays a role in both cell migration and proliferation and potentially in other physiological processes.

In summary, we have shown that TCR signaling via WNK1, OXSR1 and STK39 leads to ion influx and subsequent water influx, potentially through AQP3, which is required for TCR signaling, T cell proliferation, and hence for T-dependent antibody responses.

## Methods

### Mice
Mice with a conditional allele of *Wnk1* containing loxP sites flanking exon 2 (*Wnk1*[tm1Clhu],*Wnk1*[fl])[12], with a deletion of exon 2 of *Wnk1* (*Wnk1*[tm1Clhu], *Wnk1*[-]), with a kinase-inactive allele of *Wnk1* (*Wnk1*[tm1Tyb], *Wnk1*[D368A])[10], with a conditional allele of *Oxsr1* containing loxP sites flanking exons 9 and 10 (*Oxsr1*[tm1.1Ssy], *Oxsr1*[fl])[55], with a tamoxifen-inducible Cre in the ROSA26 locus (*Gt(ROSA)26Sor*[tm1(Cre/ESR1)Thl], *ROSA26*[CreERT2], RCE)[13], with an allele of *Stk39* encoding STK39-T243A (*Stk39*[tm1.1Arte], *Stk39*[T243A])[16] that cannot be activated by WNK1, deficient for RAG1 (*Rag1*[tm1Mom], *Rag1*[-], JAX stock #002216)[56] or SLC12A2 (*Slc12a2*[tm1Ges], *Slc12a*[-], JAX stock #034262)[57] or TCRα (*Tcra*[tm1Mjo], *Tcra*[-], JAX stock #007848)[58], expressing a tdTomato protein from the ROSA26 locus (*Gt(ROSA)26Sor*[tm4(ACTB-tdTomato-EGFP)Luo], *ROSA26*-mTmG, JAX stock #007676)[59] and with transgenes encoding TCRα and TCRβ chains specific for chicken ovalbumin residues 323-339 (Tg[(TcraTcrb)425Cbn], OT-II, JAX stock #004194)[14] have been described before. Mice with the *Aqp3*[tm2a(EUCOMM)Wtsi] allele (*Aqp3*[tm2a])[21] were imported from the European Mouse Mutant Archive (EMMA stock #08279). C57BL/6 J and B6SJLF1 mice were provided by the Biological Research Facility of the Francis Crick Institute. All genetically modified mice were maintained on a C57BL/6 J background, housed in specific pathogen-free conditions in a 12 h/12 h light/dark cycle, and given food and water *ad libitum*. Both female and male mice were used, always matching numbers of each sex between different experimental groups. All mice were at least 8 weeks old and were euthanized by cervical dislocation. All experiments were approved by the Francis Crick Institute Animal Welfare Ethical Review Board and were carried out under the authority of a Project Licence granted by the UK Home Office.

### Bone marrow chimeras
RAG1-deficient recipient mice were irradiated with 1 dose of 5 Gy from a [37]Cs source. Bone marrow cells from *Wnk1*[fl/+]RCE, *Wnk1*[fl/fl]RCE, *Wnk1*[fl/D368A]RCE, *Wnk1*[fl/+]RCE OT-II, *Wnk1*[fl/fl]RCE OT-II, *Oxsr1*[+/+]*Stk39*[+/+]RCE, *Oxsr1*[fl/fl]*Stk39*[T243A/T243A]RCE or *Tcra*[-/-] mice or fetal livers from *Slc12a2*[+/+] or *Slc12a2*[-/-] E14.5 embryos were transferred intravenously into irradiated recipient mice which were maintained on 0.02% enrofloxacin (Baytril, Bayer Healthcare) for 4 weeks. Both male and female mice were used, but in all cases the sex of the donor and recipient mice was matched. The hematopoietic compartment was allowed to reconstitute for at least 8 weeks after bone marrow transfer. In all experiments, control and experimental chimeric mice or cells derived from them were always age- and sex-matched.

### In vivo deletion of floxed alleles
Mice were injected intraperitoneally with 2 mg tamoxifen (Sigma-Aldrich) in 100 μL of corn oil (Sigma-Aldrich) on 3 (*Wnk1*[fl]) or 5 (*Oxsr1*[fl]) consecutive days. Mice were sacrificed 7 (*Wnk1*[fl]) or 21 (*Oxsr1*[fl]) days after initial tamoxifen injection.

### NP-CGG immunization and ELISA
Mice were injected intraperitoneally with 50 μg of 4-hydroxy-3-nitrophenylacetyl (NP)-CGG (BioSearch Technologies) in PBS, 25% Alum (Thermo Fisher Scientific). Blood was collected from the lateral tail vein at the indicated time points, allowed to clot, then centrifuged at 17,000 x $g$ for 10 min. $NP_{20}$-BSA (bovine serum albumin) (5μg/ml, Santa Cruz Biotechnology) was used to coat 96-well Maxisorp Immunoplates (Nalge Nunc International Corporation) overnight at 4°C. Plates were washed with PBS, 0.01% Tween-20 (PBS-T), blocked with 3% BSA in PBS, and washed with PBS, 0.01% Tween-20. Serial dilutions of sera were incubated on the $NP_{20}$-BSA-coated plates overnight at 4°C. Plates were washed 3 times with PBS-T and incubated with 1.6 μg/ml biotin-conjugated goat anti-mouse IgM (Thermo Fisher Scientific) or 4 μg/ml biotin-conjugated goat anti-mouse IgG1 (Thermo Fisher Scientific) for 2 h at room temperature. Plates were washed with PBS-T 3 times, incubated with 1 μg/ml peroxidase-labelled streptavidin (Vector laboratories) for 2 h at room temperature, washed 5 times with PBS-T and incubated with 1x TMB ELISA substrate solution (Thermo Fisher Scientific). The reaction was stopped with 2 N $H_2SO_4$. Absorption at 450 nm was measured using the SpectraMax 190 (Molecular Devices LLC.). Serum concentrations of NP-specific IgG and IgM antibodies were calculated in arbitrary units (a.u.) from the linear section of the response curves.

### Cell isolation
Blood was harvested from live mice by cutting the lateral tail vein. Peripheral lymph nodes, mesenteric lymph nodes and spleens were dissociated into single cell suspensions by mashing through a 70 μm filter. Bone marrow from the hind limbs was harvested by flushing femurs and tibias with media. Blood, bone marrow, and spleen samples were treated with ACK buffer (155 mM $NH_4Cl$, 10 mM $KHCO_3$, 0.1 mM $Na_2EDTA$) for 3 min at room temperature to remove erythrocytes, washed and resuspended in PBS. ACK treatment was repeated twice for blood samples. To isolate naive CD4+ T cells, lymph node cells were resuspended in a mixture of biotinylated antibodies against CD8, CD11b, CD11c, B220, CD19, CD25, CD44, and GR-1 (Supplementary Table 1) in IMDM (35.93 mM NaCl, 0.18 mM Penicillin, 0.17 mM Streptomycin, IMDM powder, pH 7.2 [Thermo Fisher Scientific]), 5% fetal calf serum (FCS) for 20 min at 4°C. Cells were incubated with streptavidin-coated Dynabeads (Thermo Fisher Scientific) in IMDM, 5% FCS for 20 min at 4°C. Cell and bead suspensions were placed on a magnet for 2 min at room temperature and the bead-depleted fraction was collected.

### CTV labelling of CD4+ T cells
Naïve CD4 + T cells were incubated in PBS with CellTrace Violet (CTV, ThermoFisher) at a concentration of 5 μM (for OT-II T cells) or 2.5 μM (for polyclonal T cells) for 10 min at 37°C. A 5-fold excess of pre-warmed RPMI+ medium (RPMI-1640, 100 μM non-essential amino acids, 20 mM HEPES [Thermo Fisher Scientific], 10% FCS, 100 U/mL Penicillin, 100 μg/mL Streptomycin, 2 mM L-glutamine, 100 μM 2-mercaptoethanol [Sigma-Aldrich]) was added to the cells for a further 7 min. Cells were then pelleted and resuspended in the appropriate media.

### RNA isolation and quantitative reverse transcription- quantitative polymerase chain reaction (RT-qPCR)
RNA was purified from at least $5 \times 10^4$ CD4+ T cells using the RNeasy Micro kit (Qiagen). cDNA was synthesized using the SuperScript™ VILO™cDNA synthesis kit (Invitrogen). Taqman™ Gene Expression Assays (Thermo Fisher Scientific) were used to measure *Wnk1* (exon boundary 1-2, Mm01184006_m1), *Hprt* (exon boundary 2-3, Mm03024075_m1), *Aqp3* (exon boundary 3-4, Mm00507977_g1), *Aqp9* (exon boundary 4-5, Mm00508097_m1) and *Aqp11* (exon boundary 2-3, Mm00613023_m1). Fluorescence was measured using a QuantStudio 3 or QuantStudio 5 qPCR machine (Thermo Fisher Scientific).

## Adoptive transfer of OT-II CD4$^+$ T cells

OT-II T cells (CD45.1$^-$CD45.2$^+$) were isolated from peripheral and mesenteric lymph nodes, labeled with CTV and $1 \times 10^6$ (for day 3 analysis) or $35 \times 10^3$ (for day 7 analysis) OT-II T cells were injected into the tail vein of B6SJLF1 mice (CD45.1$^+$CD45.2$^+$). 24 h later (day 0) mice were immunized with 50 μg (day 3 analysis) or 100 μg (day 7 analysis) chicken ovalbumin (OVA) protein (InvivoGen) in PBS, 25% Alum (Thermo Fisher Scientific) by intraperitoneal injection. Peripheral and mesenteric lymph nodes were harvested at day 3. Mesenteric lymph nodes, spleen and blood were harvested at day 7.

## Co-culture of OT-II T cells with APCs

To generate APCs, bone marrow from C57BL/6 J mice was treated with ACK and cultured in DC medium (RPMI-1640 [Thermo Fisher Scientific], 10% FCS, 100 U/mL Penicillin, 100 μg/mL Streptomycin, 2 mM L-glutamine [Sigma-Aldrich]) with GM-CSF (20 ng/ml, Peprotech) at $10^6$ cells/ml for 10 d at 37°C. GM-CSF was replenished at days 3, 5 and 7. Naïve CD4$^+$ OT-II T cells and APCs (both at $2.5 \times 10^5$ cells/ml) were co-cultured in round-bottomed 96-well plates in DC medium (RPMI-1640, 10% FCS, 100 U/mL Penicillin, 100 μg/mL Streptomycin, 2 mM L-glutamine [Sigma-Aldrich]) the presence of indicated concentrations of OVA$_{323-339}$ peptide (Sigma) for the indicated times.

## Activation of T cells with plate-bound anti-CD3ε and anti-CD28 antibodies

Flat-bottomed multi-well cell culture plates (Corning) were incubated overnight at 4°C with PBS containing anti-CD3ε and anti-CD28 antibodies (4 μg/ml). Naïve CD4$^+$ T cells from peripheral and mesenteric lymph nodes were cultured at $1 \times 10^6$ cells/ml in RPMI+ at 37°C. Where indicated, cells were activated in the presence of 2.5 μM WNKi (WNK463, MedChem Express), 20 μM AQP3i (DFP00173, Axon MedChem), 20 μM SLC12A2i (bumetanide, Abcam) or 10 μM SLC12A6/7i (DIOA, Sigma) which were added from a stock solution in DMSO or the equivalent amount of DMSO as a vehicle only control. 1 μM ATRi (AZD6738, Selleckchem) or 4 mM hydroxyurea (HU, Sigma-Aldrich) were added for the final 36 h or 2 h of culture, respectively. For 1 h stimulations, naïve T cells were treated with 2.5 μM WNKi, 1μM MEKi (PD0325901, Selleckchem), 20 μM SLC12A2i, 10 μM SLC12A6/7i or vehicle only and transferred into anti-CD3ε/anti-CD28-coated multi-well plates, centrifuged at 340 x g for 30 s and incubated at 37°C for 1 h, still in the presence of inhibitors. To culture cells at different osmolarities, the concentration of NaCl in RPMI+ medium was adjusted and the osmolarity verified using an osmometer (Model 3250, Thermo Fisher Scientific). In some cases, isotonic (292 mOsm/l) RPMI+ medium was used in which some NaCl was replaced with L-glucose (G5500-5G, Sigma-Aldrich), an inert osmolyte.

## Activation of T cells with soluble anti-CD3ε and anti-CD28 antibodies

Naïve CD4$^+$ T cells were incubated in IMDM (Thermo Fisher Scientific), 0.5% FCS for 1 h at 37°C, cooled at 4°C for 10 min and then incubated with anti-CD3ε and anti-CD28 antibodies (10 μg/ml) in RPMI, 0.5% FCS in the presence of 2.5 μM WNKi, 100 nM SRCi (Dasatinib, Sigma-Aldrich) or vehicle only at 4°C for 30 min. Unstimulated (0 min) samples were incubated in the presence of relevant inhibitors but without anti-CD3ε or anti-CD28 antibodies. Cells were warmed to 37°C for 15 min and then stimulated with the addition of 80 μg/mL anti-Armenian hamster IgG (Antibodies-online.com). Activation was stopped at the indicated time point with the addition of 2% paraformaldehyde (PFA).

## K$^+$ flux measurement

The rate of K$^+$ flux was measured using the FluxOR™ Potassium Ion Channel Assay (ThermoFisher) adapted to measure NKCC (SLC12A1, SLC12A2) chloride-dependent ion transport activity[60]. Isolated naive

CD4$^+$ T cells were loaded with FluxOR dye according to manufacturer's instructions, resuspended in an isotonic low Cl$^-$ assay buffer (135 mM Na Gluconate, 1 mM MgCl$_2$, 1 mM Na$_2$SO$_4$, 1 mM CaCl$_2$, 15 mM Na HEPES, 3 mM Probenecid, pH 7.4) and incubated in the presence of WNK463 or DMSO for 30 min at 4°C. Approximately $10^5$ cells were plated in 80 μl of low Cl$^-$ assay buffer per well of a 96-well plate and warmed to 37°C for 5 min. 20 μl of 500 μM Tl$_2$SO$_4$ (final concentration 100 μM) was added to all wells simultaneously and fluorescence was measured every 5.5 s for 2 min using a Spark plate reader (Tecan) with excitation/emission set to 485 nm/535 nm bandwidth 20 nm. 10 μl high Cl$^-$ buffer (135 mM NaCl, 1 mM MgCl$_2$, 1 mM Na$_2$SO$_4$, 1 mM CaCl$_2$, 15 mM Na HEPES, 3 mM Probenecid, pH 7.4) with or without Dynabeads Mouse T cell Activator CD3/CD28 beads (Gibco, 10:1 beads:cells) was added to each well and fluorescence measured for a further 9 min. All fluorescence measurements were carried out with continuous orbital shaking at 120 rpm at 37°C. Isotonicity (290-305 mOsm/L) was confirmed for all buffers with an Osmomat 3000-M (Gonotec).

The rate of chloride-dependent ion influx was determined using linear regression to determine the rate of fluorescence increase during 2 min following addition of Tl$_2$SO$_4$ and 5 min following addition of the high Cl$^-$ buffer, and normalizing the latter to the former. A single outlier was identified using the ROUT method (Q = 2%, Prism 10, GraphPad) and removed from the analysis. Finally, all measurements were normalized to the chloride-dependent ion influx in unstimulated cells in the absence of inhibitor, prior to pooling data from multiple experiments.

## Hypertonic stimulation

Isolated CD4$^+$ T cells in isotonic RPMI were centrifuged and resuspended in the same isotonic medium (300 mOsm/L) or in medium made hypertonic by adding L-glucose to RPMI to reach the desired osmolarity (350 or 400 mOsm/L). Cells were cultured with or without plate-bound anti-CD3ε and anti-CD28 antibodies for 1 h or 2 days and then analyzed for p-ERK or Ki67 respectively as described above. To measure cell division, cells were labelled with Cell Trace Violet, activated for 2 days and analyzed as described above.

## NaCl reduction and Cl replacement

To generate media of varying NaCl compositions, NaCl-free RPMI (Honeywell, S9888) was supplemented with additional NaCl. To make the medium isotonic, the remaining difference in osmolarity (up to 300 mOsm/L) was made up using L-glucose, an inert osmolyte. Osmolarity was verified using an osmometer (Model 3250, Advanced Instruments). To generate medium with reduced Cl concentration, NaCl-free RPMI was made isotonic through the addition of Na gluconate (Sigma, S2054-100G).

## Cell volume measurement

Naïve CD4$^+$ T cells were rested overnight in RPMI+ with 1 mM pyruvate. Samples were enriched for live cells by incubating in PBS at 37°C for 10 min followed by 7 min at 37°C in RPMI+ with 1 mM pyruvate. Cells were then treated with vehicle only or 2.5 μM WNKi and placed in cell culture plates ± plate-bound anti-CD3ε and anti-CD28 for 1 h at 37°C. Cell volume was measured using the CASY® cell counter (OMNI Life Science).

Alternatively, cell volume was determined using microscopy. Naïve CD4 T cells purified from ROSA26-mTmG mice expressing tdTomato-CAAX to visualize the plasma membrane were placed in glass 8-well μ-slides (ibidi, 80827) coated with either anti-CD3ε and anti-CD28 antibodies (4 μg/ml) or poly-D-lysine (50 μg/mL, Sigma, A-003-M) and incubated at 37 °C for 1 h. Cells were imaged using instant structural illumination microscopy (iSIM) using an iSIM microscope (VisiTech) with an Olympus 150x TIRF apo (1.45 NA) oil-immersion objective. Cells were excited using 552 nm wavelength laser, with emission detected using an sCMOS camera (Prime BSI Express,

Teledyne Photometrics). Resulting images were deconvoluted using Microvolution algorithms in μ-Manager. Cell volume was calculated by generating a mask of plasma membrane edges and summing up the areas of individual z-slices in each cell measured using the 'analyze particles' function in FIJI.

## Flow cytometry

**Proliferation analysis.** Cells were stained in PBS containing fluorophore-conjugated antibodies (Supplementary Table 1) and LIVE/DEAD™ Near-IR dye (Thermo Fisher Scientific) (1:500) at 4°C for 20 min, and resuspended in FACS buffer (PBS, 0.5% BSA). For cell counting, a known number of Calibrate APC beads (BD biosciences) were added to the sample. Samples were analyzed on an LSRFortessa™ (BD biosciences) flow cytometer. Flow cytometry data were analyzed using FlowJo 9 and 10 (BD Biosciences).

**CXCR5.** Cells were incubated in PBS, 2% FCS containing biotinylated anti-CXCR5 antibody at 37°C for 30 min. Cells were washed, incubated with fluorophore-conjugated streptavidin for 15 min at room temperature, washed, resuspended in FACS buffer, and analyzed on the flow cytometer.

**Analysis of phosphorylated proteins.** Cells were fixed in 2% PFA at 4°C for 15 min, permeabilized with 90% methanol overnight at -20°C, washed 3 times and then incubated with the relevant primary antibody (Supplementary Table 1) in PBS, 2% FCS for 30 min at room temperature. Cells were then washed, stained with the relevant secondary antibody in PBS, 2% FCS for 30 min at room temperature, washed, resuspended in FACS buffer, and analyzed on the flow cytometer.

**Cell cycle analysis.** To analyze the rate of S phase progression, 10 μM EdU (5-ethynyl-2'-deoxyuridine, Jena Bioscience) was added to the cell culture medium for 10 min at the end of 72 h culture on anti-CD3ε- and anti-CD28-coated cell culture plates. Cells were fixed in 4% PFA, permeabilized in BD Perm/Wash™ buffer (BD biosciences), 1% BSA at room temperature for 15 min. The Click reaction was initiated by adding 2 mM CuSO$_4$, 0.3125 μM AF488-azide, 10 mM Sodium Ascorbate (all from Jena Bioscience) in TBS (50 mM Tris, 150 mM NaCl, pH 7.5) for 30 min at room temperature. To analyze distribution of cells in different phases of the cell cycle, cells were stained with Live/Dead-Near IR stain (ThermoFisher Scientific) and fixed in 4% PFA for 15 min at 4°C, stained with fluorophore-conjugated antibodies against Ki67, p-MPM2 (Supplementary Table 1) in BD Perm/Wash™ buffer (BD biosciences) containing 2% FCS for 30 min at room temperature, resuspended in BD Perm/Wash™ buffer containing 1 μg/mL FxCycle™ Violet (Thermo Fisher Scientific) to stain for DNA content, incubated at room temperature for 30 min, and analyzed on the flow cytometer. In some experiments, cells were pulsed with EdU and analyzed as above. Ki67 was used to identify cells that had exited G0. The proportion of cells in G1, S, and G2/M was estimated using the Watson model generated from the histogram of DNA content of Ki67$^+$ cells. Staining for p-MPM2 was used to identify cells in M phase. Where indicated, cells were pulsed for 30 min with EdU at the end of 72 h culture and G2/M cells were identified as the 4n EdU$^-$ population. The effect of expansion was taken out of the data by dividing the total number of Ki67$^+$ cells by the expansion index - the fold expansion of the population due to proliferation estimated by generating a model using CTV dilution histograms – to give Ki67$^+_{adjust}$. The percentage of Ki67$^+$ cells in the population if no division had taken place was then calculated as follows: (Ki67$^+_{adjust}$)/(Ki67$^+_{adjust}$ + Ki67$^-$) x 100%.

**Membrane potential.** Isolated CD4$^+$ T cells were incubated in RPMI medium containing 30 μM Cytovolt2 dye (Cytocybernetics) for 2 min, then returned to dye-free isotonic medium (297 mOsm/L), NaCl-free medium supplemented with either L-glucose or with Na gluconate to make it isotonic, or hypotonic medium with reduced NaCl (165 mOsm/L). Fluorescence of the dye was acquired by flow cytometry for 1 min. Gramicidin (final concentration 5 mg/ml, Sigma Aldrich) was added to the sample and fluorescence acquired for a further 3 min.

## Immunoblot analysis

Cell pellets were thawed at room temperature, lysed using RIPA buffer (50 mM Tris, 150 mM NaCl, 50 mM NaF, 2 mM EDTA, 2 mM Na$_4$P$_2$O$_7$, 1 mM Na$_3$VO4, 1% Triton X-100, 0.5% deoxycholate, 0.1% SDS, 1 mM PMSF, cOmplete™ EDTA-free protease inhibitor cocktail [Roche], PhosSTOP™ [Roche]), centrifuged at 17,000 x $g$ for 15 min at 4°C and the supernatant was collected. To measure the effect of hypertonic medium, cells in isotonic RPMI were warmed to 37°C for 5 min and pre-warmed hypertonic medium (RPMI with L-glucose added to 700 mOsm/L) was added to make a final osmolarity of 400 mOsm/L, incubated for a further 5 min and lysed with an equal volume of 2x RIPA buffer. The protein concentration of each sample was determined using the Pierce™ BCA protein assay kit (Thermo Fisher Scientific) according to the manufacturer's instructions. Samples were mixed with NuPage™ LDS sample buffer (Thermo Fisher Scientific), 50 mM DTT and loaded onto NuPAGE™ 4-12% Bis-Tris precast polyacrylamide gels (Thermo Fisher Scientific) with the volume of each sample adjusted to ensure equal protein amounts per well. Electrophoresis was carried out at 200 V for 53 min in NuPAGE™ MOPS or MES SDS running buffer (Thermo Fisher Scientific). Proteins were transferred to a methanol-activated polyvinylidene difluoride membrane in NuPAGE™ transfer buffer (Thermo Fisher Scientific) at 30 V for 65 min. Membranes were blocked with Odyssey® blocking buffer (LI-COR) for 1 h at room temperature, incubated with primary antibody (Supplementary Table 1) in Odyssey® blocking buffer overnight at 4 °C, washed 3 times with TBS-T (TBS, 0.05% Tween20), incubated with fluorophore-conjugated secondary antibody (Supplementary Table 1) in Odyssey® blocking buffer for 1 h at room temperature, washed 3 times with TBS-T, and imaged using the Odyssey® CLx (LI-COR). Image Studio Lite (LI-COR) was used to quantify intensities of bands. Full immunoblots are shown in the Source Data file.

## RNA sequencing (RNAseq)

*Wnk1$^{+/}$*RCE and *Wnk1$^{-/}$*RCE CD4$^+$ T cells were purified from lymph nodes of bone marrow radiation chimeras that had been treated with tamoxifen as described above. Cells were lysed using QIAzol lysis kit (Qiagen) and RNA was extracted using RNAeasy mini kit (Qiagen) according to the manufacturer's instructions. cDNA library synthesis was carried out using KAPA mRNA HyperPrep Kit (Illumina) and libraries were sequenced on the Illumina HiSeq 4000, acquiring 75 base single end reads. Reads were trimmed and adaptors removed using Trimmomatic (v 0.36). Reads were aligned with STAR (v 2.5.2a) using mouse genome assembly GRCm38, gene counts quantified with RSEM (v 1.2.31) and normalized counts generated using DESeq2[61]. Expression values for all genes and samples is available in Supplementary Data 1. Data has been deposited with GEO (accession number GSE201228 https://www.ncbi.nlm.nih.gov/geo/query/acc.cgi?acc=GSE201228).

## Water permeability

Stopped-flow experiments were conducted at 25 °C on a HiTech SF61 DX2 apparatus, equipped with a mercury–xenon lamp (TgK Scientific Ltd). Suspensions of naive CD4$^+$ T cells (6×10$^6$/ml in PBS) were rapidly mixed with an equal volume of 300 mM mannitol in PBS to give an inwardly directed osmotic gradient of 150 mM. In the inhibitor experiments 10 μM DFP00173 (AQP3i) was included in the solutions and was incubated for at least 15 min at room temperature before the measurements. Time courses of 90° light scattering were recorded at 450 nm with a 12 nm bandwidth over 60 s. At least four traces were measured for each biological replicate and averaged before data

analysis. Scattering time traces were analysed in Kinetic Studio 5.0 (TgK Scientific Ltd) by fitting a single exponential curve to the data to determine the observed rate constant. Data from each batch of T cells were then normalized to the average of the rate constant obtained in the absence of inhibitor (relative rates).

### Catalase treatment to eliminate $H_2O_2$

To establish the concentration of catalase required to eliminate $H_2O_2$, mouse $CD4^+$ T cells were loaded with DCFDA (ab113851, Abcam) according to the manufacturer's protocol. Cells were pre-treated with 0, 2, 20, 200 U/ml bovine liver catalase (C40, Sigma Aldrich) for 30 min. Cells were treated with 1 mM $H_2O_2$ for 30 min at room temperature, and DCFDA fluorescence was measured by flow cytometry (LSRFortessa X-20, BD Biosciences). To measure the effect of eliminating $H_2O_2$ on TCR/CD28-induced ERK activation, $CD4^+$ T cells were incubated with 0, 2, 20, 200 U/ml of catalase at 37 °C for 30 min before transferring to anti-CD3ε, anti-CD28 precoated plates and incubated for 1 h. Cells were fixed, permeabilized and stained for pERK as described earlier.

### Statistical analysis

The non-parametric Mann-Whitney U-test, Kruskal-Wallis test or a 2-way ANOVA were used to analyze the difference between groups using GraphPad Prism 8. All statistical tests were 2-sided and always derived from biological not technical replicates.

### Reporting summary

Further information on research design is available in the Nature Portfolio Reporting Summary linked to this article.

## Data availability

Mouse strains available on request. RNAseq data has been deposited with GEO (accession number GSE201228 https://www.ncbi.nlm.nih.gov/geo/query/acc.cgi?acc=GSE201228). Source data are provided with this paper.

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

## Acknowledgements
We thank Erik Sahai and Simon Boulton for critical reading of the manuscript. We thank Matt Renshaw for help with image analysis and the Structural Biology Facility of the Francis Crick Institute for help with stopped flow. We thank Souradeep Basu, Subramanian Venkatesan, Daisy Luff, Darryl Hayward, Edina Schweighoffer, Simon Boulton, and Rob de Bruin for advice. We thank the Flow Cytometry, Advanced Light Microscopy and Biological Research Facilities of the Francis Crick Institute for flow cytometry, imaging and animal husbandry. We thank Chou-Long Huang, Dario Alessi and Sung-Sen Yang for mouse strains. VLJT was supported by the Francis Crick Institute which receives its core funding from Cancer Research UK (CC2080), the UK Medical Research Council (CC2080), and the Wellcome Trust (CC2080), and by a grant from UKRI Biotechnology and Biological Sciences Research Council (BB/V0088757/1). LLdB was funded by an Imperial College London President's PhD Scholarship. For the purpose of Open Access, the author has applied a CC-BY public copyright licence to any Author Accepted Manuscript version arising from this submission.

## Author contributions
Conceptualization: J.B.O.M., V.L.J.T. Data curation: J.B.O.M. Formal analysis: J.B.O.M., L.dB., M.L. Funding acquisition: V.L.J.T. Investigation: J.B.O.M., L.V., L.dB., H.H., D.H., R.K., S.K., D.L. Methodology: J.B.O.M., R.K. Project administration: V.L.J.T. Resources: R.K. Supervision: V.L.J.T. Visualization: J.B.O.M. Writing – original draft: J.B.O.M., V.L.J.T. Writing – review & editing: J.B.O.M., V.L.J.T.

## Funding

## Competing interests
The authors declare no competing interests.
