## [Transparent Peer Review file · Nature Communications]

WNK1-dependent water influx is required for CD4⁺ T cell activation and T cell-dependent antibody responses

Corresponding Author: Professor Victor Tybulewicz

Version 0:

Reviewer comments:

Reviewer #1

(Remarks to the Author)

In this study, O'May et al. investigate the WNK1-OXSR1-STK39 pathway in T cells, and show that its activation after TCR stimulation is required for CD4 T cell proliferation, cell cycle regulation and antibody responses. Moreover, they show that activation of the pathways leads to water influx through AQP3, presumably by WNK1 dependent phosphorylation of Na/K/Cl transporters of the SLC12 family. The authors conclude that regulation of water influx by WNK1 may be a common feature of mitogenic pathways in many cell types". The authors use a variety of conditional knockout (cKO) mice deleting WNK1, OXSR1 and STK39 genes and inhibitors of these genes and AQP3. Overall the experiments are well conducted and the conclusions justified. The described pathway builds on an earlier paper by the same lab, but is still novel in the context of T cells and immune responses. It is significant because it links T cell activation to regulatory volume increase (RVI) and cell cycle progression, which is a prerequisite of clonal expansion and effective immune responses. Notwithstanding my general excitement about this study, some experiments need to be conducted more rigorously to support the authors conclusions. The paper also lacks a mechanistically satisfying explanation of how WNK1-OXSR1-STK39 signaling regulates SLC12 and AQP3 and thus cell volume.

Specific comments:

Figures 1 and 2 use cKO mice to demonstrate the importance of the WNK1-OXSR1-STK39 pathway for T-dependent antibody responses. These experiments are generally well done and convincing, but I have several comments. 1) Why did the authors not use *Wnk1* floxed mice crossed to *Cd4Cre* or *Cd4-ERT2Cre* to conduct their experiments, and instead use radiation chimeras, which is a less clean approach (associated with potential side effects of irradiation)? 2) I disagree with the sentence on page 5 (regarding Figure 1D-J) "... very few [WNK1-deficient OT-II T cells] had differentiated into TFH cells", which is also mentioned in the abstract "WNK1-deficient CD4⁺ T cells are severely impaired in their ability to proliferate and differentiate into T follicular helper cells in response to immunization with a T-dependent antigen." I agree that WNK1-deficient CD4 T cells fail to proliferate, but they do not have a selective defect in Tfh cell differentiation. Without proliferation, they very likely cannot differentiate into Th1, Th2 or other T cell subsets either. I suggest to rephrase this conclusion. Minor comment: The IgG1 data shown in Figure 1B seem to be very variable, please show individual data points.

In the second half of the paper (Figures 3-5), the authors rely on pharmacological inhibitors of the WNK1-OXSR1-STK39 pathway, AQP3 and other molecules instead of making use of their cKO mice or CRISPR gene editing. Some of the findings regarding the role of AQP3 in cell cycle regulation or T cell function should be confirmed genetically (for instance Figure 3K,M, but also Figures 4 and 5).

Figure 4. Please show primary data for CD3, ZAP70 and ERK phosphorylation as a supplementary figure. Are the MFI for P-CD3, P-ZAP70 and P-ERK normalized to total protein MFI and to cell size (FCS)? This is important because WNK1 inhibition reduces cell size as the authors showed, and this may cause an apparent reduction of P-ERK MFI, which may be an artifact.

I have a problem with the conclusion that the WNK1-OXSR1-STK39 pathway regulates cell volume via activation of SLC12 transporters and AQP3. This is an attractive model, which the authors show in Figure 5, but it relies on few experiments and many assumptions:

1) The conclusion that the WNK1-OXSR1-STK39 pathway regulates cell volume rests entirely on the fact that a WNK inhibitor reduces cell volume and that AQP3 inhibition impairs proliferation. How was cell volume measured? The methods section is vague about it. Because this is a central argument in the paper, this aspect needs to be investigated in more detail. Dynamic changes in cell volume can be measured, for instance, using dyes such as calcein.

2) The model in Figure 5 shows SLC12A2 (and the introduction specifically mentions it), but there are no data to support its role. The authors argue (page 7) that it is "impractical" to delete all four SLC12 genes that are expressed in T cells, but they have shown before that SLC12A2 is involved in T cell migration. I would therefore argue that they need to delete SLC12A2 in T cells and measure T cell function, cell cycle and RVI.

On page 12, the authors conclude that WNK1 deletion results in ATR activation independently of DNA damage. What is the evidence for that?

An interesting question to ask would be whether impaired RVI in WNK1 deficient T cells can be rescued (and thus impaired T cell proliferation) by blocking RVD (regulatory volume decrease) mediated by LRRC8 channels using CRISPR or DCPIB. This is not necessary but something for the authors to consider.

Reviewer #2

(Remarks to the Author)
Comments to the Authors

This manuscript by O'may colleagues explored whether WNK1 is responsible for CD4+ T cell activation and T cell-dependent antibody responses. O'may et al. provide a comprehensive and detailed analysis of WNK1 downstream effector signaling pathways regulating T cell-dependent antibody responses. They found that OXSR1, STK39, and AQP3 is a downstream signaling cascade of WNK1 corresponding T cell activation. They further demonstrate that WNK1 pathway-stimulated AQP3-driven water influx is vital for TCR-mediated proliferation and cell-cycle regulation. Overall, the manuscript is well written in general, and the contents are convincing for most of the parts. However as discussed below, this study is missing several points and need to be interpreted with care.

Major Concerns:

1. In the former author's study, CCL21-induced CCR7 activation mediated cell migration to activate CD4+ T cells via the WNK1-OXSR1-STK39-SLC12A2 pathway. The current study demonstrated that the same downstream signaling cascade is critical for TCR/CD28-induced activation of CD4+ T cells independently of regulation of migration and adhesion. Although the activation of CCR7 and TCR shared their downstream effector signaling pathway, the physiological outcomes (migration vs proliferation) seem to be different. What's the reason or underlying mechanism for these differences?
2. Authors proposed that WNK1-deficient T cells were markedly impaired in TCR/CD28-induced proliferation via the AQP3-dependent pathway. In Fig 2, the AQP3i-induced reduction of TCR/CD28-activated proliferation was less than that seen in the WNK1 deletion. This raises the question that WNK1 may regulate T cell proliferation not only via the AQP3-driven pathway but also by alternative pathways.
3. Current study provides evidence that the WNK1 regulates T cell proliferation and cell cycle via OXSR1/STK39/AQP3-mediated water influx. Is this an exclusive pathway or mechanism for WNK1-mediated cell-cycle regulation? OXSR1 is a well-characterized WNK1 downstream target kinase and is activated in response to osmotic stress by WNK family members. Melanie Cobb's lab (Tu et al., PNAS 2011) reported that WNK1 regulates mitosis via the OXSR1-independent pathway. In addition, the same group recently reported WNK1 as an autophagy regulator (Gallolu Kankanamalage, Autophagy 2017; Gallolu Kankanamalage, PNAS 2016). Indeed, autophagy pathways are tightly linked with cell-cycle progression. Key experiments to exclude those alternatives would strengthen the conclusions drawn.
4. There are clear genetic links between WNK1, OXSR1, and STK39 and T cell-dependent immune response. Is there any genetic relationship between AQP3 and WNK1 or AQP3 and altered T cell-dependent immune response? Although the authors used a pharmacological AQP3 inhibitor (AQP3i) to block AQP3, the loss of function experiments such as CRISPR knockout would be required to gather convincing evidence to support their conclusions.
5. Authors proposed that WNK1-activated AQP3-driven water influx is the primary mechanism for CD4+ T cell proliferation mediating T cell-dependent antibody response. The authors demonstrated a mechanism that WNK1 stimulates AQP3-driven water influx through the OXSR1-STK39-SLC12A2 pathway. However, several alternative mechanisms can be proposed. It is well established that WNK1 regulates cell surface expression of multiple channels and transporters via regulating exo- and/or endocytosis of them. WNK1 can directly activate AQP3 by phosphorylation and/or physical protein interaction. Hence, it is conceivable that WNK1 may upregulate AQP3 by activating intrinsic channel properties directly (activity, etc.) or by regulating the cell-surface abundance of the AQP3. The authors need to carefully answer raised questions.
6. It is known that AQP3 is permeable to glycerol and H₂O₂ as well as water. Glycerol and H₂O₂ uptake via AQPs are known to be required for T-cell migration and activation. The current study suggested that AQP3-driven water influx is responsible for CD4+ T cell proliferation and antibody response. Is there any chance that AQP3-mediated glycerol and/or H₂O₂ transport can contribute to regulating T cell differentiation and activation?
7. In Fig. 5, hypotonic stress rescued WNK1 mutant- or WNK1i-blunted anti-CD3/CD28 response through water influx, whereas the isotonic medium with reduced Na/Cl concentrations failed to reverse it. Hence, the authors concluded that the water influx but not ion concentration is responsible for T cell activation. The authors examined isotonic medium containing

low concentration NaCl. It claims several concerns. Lower external Cl⁻ concentration can shift Cl⁻ equilibrium potential in a more positive direction. It may elicit inward Cl⁻ current (Cl⁻ efflux) at RMP leading to water efflux. It is also reported that hypotonic stress promotes vascular cell proliferation via WNK1-mediated volume-regulated Cl⁻-channel (VRCC) regulation. Thus, the current study does not fully exclude the possibility that hypotonic stress regulates TD antibody response through WNK1-mediated ion channel regulation such as VRCC and Ca²⁺-activated Cl⁻ channel. Additionally, it is well established that hypertonic stress activates WNK1 phosphorylation. Does hypertonic stress mimic the TCR-induced proliferation of CD4⁺ T cells?

8. Deletion of one of the WNK families can lead to compensatory expression of another WNK isoform. For example, WNK1 was markedly overexpressed in WNK3 knockout mice (Mederle et al., *Am J Renal Physiol* 2013). Although the author's former study identified that Wnk1 is the only member of the WNK family expressed in thymocytes, it would be important to assess whether other WNK family is upregulated in WNK1-deficient CD4⁺ cells to compensate.

9. Authors showed that WNK1 deficiency disturbed T cell proliferation and TD antibody responses. Can forced WNK1 overexpression in WNK1-deficient T cells rescue the phenotypes? Additionally, the gain of function experiment would be supporting the notion that WNK1 is required for CD4⁺ T cell activation.

10. The authors demonstrated that WNK1 kinase activity is required for the AQP3 activations using a kinase-inactive allele of WNK1 (Wnk1D368A). Apart from D368A, do other kinase active sites also affect CD4⁺ cell proliferation?

Reviewer #3

(Remarks to the Author)

In this study, Biggs O'May et al. demonstrated that water influx via AQP3 mediated by WNK1-OXSR1/STK39 signal increased the proliferation of CD4-positive T cells. Moreover, they showed that WNK1-deficient CD4-positive T cells were impaired in the differentiation into T follicular helper cells and subsequent T cell-dependent antibody responses. Regarding to the mechanism, they demonstrated that water influx mediated by the WNK1 pathway activated ERK signaling and promoted entry into G1 phase in the cell cycle. There are several suggestions for improvement.

1. The authors used WNK1fl/- RCE mice to delete WNK1 specifically in CD4-positive T cells. The other immune cells including B cells should also be checked for WNK1 expression as well as CD4-positive T cells.
2. In Figure 1A, the chimerism of blood cells should be checked to confirm whether the proliferation of WNK1-deficient CD4-positive T cells is suppressed and whether other blood cells are affected. Is it possible to confirm the chimerism of blood cells using CD45.1 positive mice or labelling fluorescent proteins such as GFP?
3. In Figure 1A, what are the results of using only WNK1fl/- RCE mice as donors without using Tcr α ^{-/-} mice? Will there be a decrease in lymphocyte count?
4. Regarding to their scheme in Figure 5, they demonstrated that SLC12A2 was involved in the mechanism. However, they didn't provide any data about SLC12A2. Phosphorylation of activation site of SLC12A2 in TCR signaling and the effect of SLC12A2 inhibition on CD4⁺ T cell proliferation needs to be assessed.
5. For Figure 2L-M, the authors should show the expression level of AQP family in CD4⁺ T cells or cite previous articles that show AQP3 is the predominant AQP in CD4⁺ T cells. Coupling of SLC12A2 and AQP3 shown in the summary figure 5D is not directly proved in this study. Can the effect of WNK signaling on the activity of AQP3 (such as trafficking to the cell membrane) be assessed?
6. Observation that reduced tonicity rescued WNK1 deficiency is interesting in Figure 5. Is this effect through only AQP3? Does WNK1/OXSR1/STK39 knockout or knock-in affect water permeability in T cell?
7. Water permeability should be examined, as in Fig. 5 of previous paper (PMID : 22927550).
8. Intracellular localization of AQP3 should be shown, as in Fig. 2d of previous paper (PMID : 22927550). Does WNK1/OXSR1/STK39 knockout or knock-in affect intracellular localization of AQP3?
9. Please show cell images (micrograph) before and after cell volume is changed. Often, CASY cell counter cannot accurately measure cell volume.
10. The authors used only an AQP3 inhibitor, DFP00173, to examine the physiological role of AQP3 and did not directly evaluate water permeability of AQP3 in T cell. Importantly, DFP00173 inhibits not only water transport but also glycerol and H₂O₂ transport (PMID: 30862673). In addition, it has been reported that T cell migration is dependent on AQP3-mediated "hydrogen peroxide (H₂O₂) uptake" but not the canonical water/glycerol transport (PMID : 22927550). Therefore, there is a possibility that WNK1/OXSR1/STK39 signaling affect H₂O₂ transport, resulting in modulation of T cell migration. Further examinations of physiological analysis of AQP3 are required in this study, to answer these questions. Experiments using AQP3-deficient (knockout) T cell would be best show the direct involvement into the proliferation of CD4-positive T cells

Minor

1. There are many abbreviations (ex. RCE) that are not spelled out in the manuscript.
2. The authors group revealed previously that WNK1 is involved in attachment and migration of CD4⁺ T cells (Köchel et al. *Nat Immunol* 2016) and recently showed the same molecular mechanism is important in migration and proliferation of B cells (Hayward et al. 2021 <https://doi.org/10.1101/2021.09.09.459588>). It was also reported previously that AQP3 is involved in keratinocyte's proliferation and migration (Hara-Chikuma et al. *J Mol Med* 2008), and T cell migration (Hara-Chikuma et al. *J Exp Med* 2012). Based on these publications, involvement of WNK1 signaling and AQP3 in CD4⁺ T cell proliferation is interesting but could have less impact. The authors must emphasize the novelty of this manuscript.
3. The authors concluded in the previous paper (Köchel et al. *Nat Immunol* 2016) that WNK1 does not contribute to ERK activation in TCR signaling. This is inconsistent with Figure 4 in the current manuscript and needs to be discussed.

Version 1:

Reviewer comments:

Reviewer #1

(Remarks to the Author)

The authors have addressed many of the previously raised concerns by conducting a number of new experiments and rewriting some sections of the manuscript. These changes have clarified some of the issues that were brought up and improved the study. Nonetheless, I am not entirely convinced of some of their conclusions and the proposed model in Figure 7D, which is not fully supported by the data.

1. The new experiments in which SLC12A2 was inhibited or deleted in T cells do not provide any evidence that SLC12A2 (or another homologue of the SLC12 family) regulates ERK phosphorylation, cell volume or water influx. The fact that SLC12A2 is phosphorylated in a WNK1 dependent manner is not sufficient to implicate it in T cell function. To be fair, the authors appropriately discuss these limitations in the text, but should also remove SLC12A2 from the model in Figure 7.
2. The experiments with the WNK inhibitor in Figure 3 show reduced ZAP70 and ERK phosphorylation after TCR stimulation. A much weaker reduction of ERK phosphorylation is observed with the AQP3 inhibitor. These data suggest that WNK1 likely has direct effects on TCR signaling that are independent of AQP3, which may explain much of the defects in T cell proliferation, cell cycle arrest and in vivo phenotype (Tfh cells etc).
3. Because of these concerns and the fact that water influx through AQP3 is a central conclusion of their study, some form of genetic deletion (CRISPR, RNAi, KO) to validate these findings is important in my opinion. As pointed out in the original critiques, the involvement of AQP3 in T cell function is solely based on the use of an inhibitor. This limitation is made worse by the fact that the molecular link between WNK1 and AQP3 is missing (as SLC12A2 has been ruled out).
4. From the experiments in Figure 5, in which NaCl concentrations in the media are reduced, the authors conclude "Na⁺ and Cl⁻ ions are likely important for TCR/CD28 signaling" because ERK phosphorylation is reduced. These experiments are used to support the idea that Na⁺ and Cl⁻ influx by SLC12A promotes subsequent water influx by AQP3, but the evidence is weak. In addition to the concerns discussed above, I see two additional problems:
 - A) the authors do not show that Na⁺ and/or Cl⁻ uptake is indeed altered in Wnk1, Oxsr1 or Stk39 deficient T cells to support the idea that Wnk1 regulates Na⁺, K⁺ or Cl⁻ levels in cells. Na⁺ levels can be measured, for instance, with fluorescent Na⁺ dyes; K⁺ can be measured by thallium flux measurements. These experiments are absolutely critical in my opinion.
 - B) Na⁺ and Cl⁻ ions may affect TCR signaling by altering the membrane potential. A negative membrane potential is essential for T cell signaling as many studies in K⁺ and Na⁺ channel deficient T cells have established. If the authors want to use the data in Figure 5A,B to support their model, they need to exclude effects on the membrane potential as a reason for the observed decrease in ERK phosphorylation.

Reviewer #2

(Remarks to the Author)

Comments to the Authors

Most comments in this revised manuscript have been addressed nicely with additional data to support their conclusions. However, several concerns still need clarification for the manuscript to be considered credible and improved.

1. In response to comment #7 from reviewer 2, my questions were regarding two aspects: 1) The effects of lowering extracellular Cl⁻ shifting its equilibrium potential eliciting inward Cl⁻ currents on WNK1 activation and its downstream events. 2) The effects of hypertonic stress on WNK1 phosphorylation and the subsequent WNK1-induced events. The authors must provide thorough answers to these raised questions for a more comprehensive understanding of their findings.
2. A key finding of this study is that TCR activation leads to water entry, which regulates CD4⁺ T cell proliferation and T-cell-dependent antibody responses through a WNK1 kinase activity-dependent mechanism. While the authors have demonstrated TCR-stimulated ERK phosphorylation and conducted a Wnk1(D368A) knock-in experiment, they have not examined WNK1 phosphorylation and expression levels. It is imperative that the authors provide this data to strengthen the evidence supporting their conclusions.
3. The authors conducted new RNA sequencing analysis in Wnk1^{+/-}RCE and Wnk1^{-/-}RCE CD4⁺ T cells and identified three detectable isoforms of AQP genes in CD4⁺ T cells through RNASeq analysis. Have any differences been observed in the expression levels of WNK1 downstream targets, including AQPs, OXSR1, STK39, and NKCCs, by Wnk1^{-/-}?
4. The authors proposed that Na⁺ and Cl⁻ ions play a significant role in TCR/CD28 signaling, independently of their effect on osmolarity. To enhance the understanding of this mechanism, the authors should elaborate on the underlying mechanism behind the role of Na⁺ and Cl⁻ ions on T cell signaling and WNK1 regulation. Providing this information will help establish a more comprehensive framework for their findings.

Overall, addressing these concerns and providing the necessary data and explanations will substantially improve the credibility and impact of the manuscript.

Reviewer #3

(Remarks to the Author)

Although inadequate in some areas, many parts of the system answer my request. However, some major comments need to be answered. In particular, I would like you to demonstrate that inhibition of WNK1 suppresses water permeability in this system.

Major:

1: Water permeability of AQPs is usually measured by stopped flow approach with osmotic gradient, since AQPs are channels but not transporters. Osmotic gradient is required to induce water transport through AQPs. In this study, the authors concluded that TCR/CD28 stimulation activated SLC12A-family of ion co-transporters, leading to the net influx of Na⁺, K⁺ and Cl⁻ ions. Inward ion flux worked as osmolytes which was driving force for increasing water permeability via AQP3. However, in Figure R1, osmotic gradient was formed by inward 150 mM mannitol, which overwhelmed small osmotic gradient formed by WNK1-OXSR1-STK39 signaling; therefore, inhibition of WNK1 did not affect water permeability. Instead of mannitol, the authors should use isotonic buffer containing Na⁺, K⁺ and Cl⁻ ions to evaluate WNK1-induced AQP3 water transport after TCR/CD28 stimulation. Furthermore, the authors should simply prove that inhibition of WNK1 reduces water permeability in this T cell system.

2: In this revision, the authors have additionally performed RNA-seq using WNK1^{+/+} RCE and WNK1^{-/-} RCE CD4⁺ T cells. Analyzing these data, can WNK1^{-/-} have suppressed any transcriptional signatures associated with TCR signaling and/or ATR-mediated G2/M checkpoint?

Version 2:

Reviewer comments:

Reviewer #1

(Remarks to the Author)

The authors have responded to my follow-up requests with detailed new experiments, which have addressed my concerns. I agree with their changes to the manuscript. Thank you and congratulations on a nice study.

Reviewer #2

(Remarks to the Author)

I have carefully reviewed the revised manuscript and commend the authors for their thorough revisions. Most of the comments from the previous review have been addressed satisfactorily, and the additional data provided significantly strengthens the study's conclusions.

The authors have done an excellent job addressing most of the issues raised in the initial review. Including additional data has provided further support for their conclusions. The explanations offered in the revised text are clear and align well with the newly added figures. The authors have appropriately expanded the discussion to incorporate the additional findings. The manuscript has significantly improved, and I believe it is now in a strong position for publication. The authors have effectively incorporated the feedback, and the additional data adds substantial value to the study. I recommend the manuscript for publication with minor revisions, if any.

Reviewer #3

(Remarks to the Author)

Fig.6a-c are convincing data that cell volume is regulated by TCR/WNK1 signaling. I have no further comments.

Response to Reviewers' Comments

Reviewer #1 (Remarks to the Author):

In this study, O'May et al. investigate the WNK1-OXSR1-STK39 pathway in T cells, and show that its activation after TCR stimulation is required for CD4 T cell proliferation, cell cycle regulation and antibody responses. Moreover, they show that activation of the pathways leads to water influx through AQP3, presumably by WNK1 dependent phosphorylation of Na/K/Cl transporters of the SLC12 family. The authors conclude that regulation of water influx by WNK1 may be a common feature of mitogenic pathways in many cell types". The authors use a variety of conditional knockout (cKO) mice deleting WNK1, OXSR1 and STK39 genes and inhibitors of these genes and AQP3. Overall the experiments are well conducted and the conclusions justified. The described pathway builds on an earlier paper by the same lab, but is still novel in the context of T cells and immune responses. It is significant because it links T cell activation to regulatory volume increase (RVI) and cell cycle progression, which is a prerequisite of clonal expansion and effective immune responses. Notwithstanding my general excitement about this study, some experiments need to be conducted more rigorously to support the authors conclusions. The paper also lacks a mechanistically satisfying explanation of how WNK1-OXSR1-STK39 signaling regulates SLC12 and AQP3 and thus cell volume.

Specific comments:

Figures 1 and 2 use cKO mice to demonstrate the importance of the WNK1-OXSR1-STK39 pathway for T-dependent antibody responses. These experiments are generally well done and convincing, but I have several comments.

1) Why did the authors not use *Wnk1* floxed mice crossed to Cd4Cre or Cd4-ERT2Cre to conduct their experiments, and instead use radiation chimeras, which is a less clean approach (associated with potential side effects of irradiation)?

We were not able to use the Cd4Cre because WNK1 is required during thymic development, so we needed to use an inducible system. The Cd4CreERT2 system would probably have been suitable for this purpose, but this is not an allele we had in house. Instead, we chose to use the ROSA26-CreERT2 allele to delete the *Wnk1* gene, since we have extensive experience using this allele, and know that we get very efficient deletion of the conditional *Wnk1* allele. Of course, as the reviewer points out, it necessitates generating mixed bone marrow chimeric mice, combining the test marrow with TCR α KO marrow and reconstituting irradiated RAG1-deficient mice. In the resulting chimeras all the $\alpha\beta$ T cells are derived from the test marrow. As the reviewer points out, irradiation can have side effects, but of course in these experiments this is controlled, since the chimeras receiving the control marrow were also irradiated. Finally, we note that to reconstitute RAG1-deficient mice, we give them a relatively low sub-lethal dose radiation (5 Gy), whose effects are likely to be mild.

2) I disagree with the sentence on page 5 (regarding Figure 1D-J) " ... very few [WNK1-deficient OT-II T cells] had differentiated into TFH cells", which is also mentioned in the abstract "WNK1-deficient CD4+ T cells are severely impaired in their ability to proliferate and differentiate into T follicular helper cells in response to immunization with a T-dependent antigen." I agree that WNK1-deficient CD4 T cells fail to proliferate, but they do not have a selective defect in

Tfh cell differentiation. Without proliferation, they very likely cannot differentiate into Th1, Th2 or other T cell subsets either. I suggest to rephrase this conclusion.

We agree with the reviewer that we do not have evidence for a specific defect in differentiation into Tfh cells. As suggested, we have changed the wording in the abstract and in the Results (page 5, paragraph 2).

Minor comment: The IgG1 data shown in Figure 1B seem to be very variable, please show individual data points.

As requested, we have edited Figure 1B to show individual data points.

In the second half of the paper (Figures 3-5), the authors rely on pharmacological inhibitors of the WNK1-OXSR1-STK39 pathway, AQP3 and other molecules instead of making use of their cKO mice or CRISPR gene editing. Some of the findings regarding the role of AQP3 in cell cycle regulation or T cell function should be confirmed genetically (for instance Figure 3K,M, but also Figures 4 and 5).

Several years ago, based on published papers, we had tried to obtain AQP3-deficient mice, but the labs we approached were unable to provide them. More recently we have identified that there is a EUCOMM allele of *Aqp3* available which can be used to generate *Aqp3* knockout mice. However, the process of importing and establishing this strain and crossing it to a germline Flp and then Cre driver to generate the knockout mice will take a substantial amount of time and is beyond the scope of what we could practically do for this response. To circumvent this, we attempted to knockout *Aqp3* in primary mouse CD4⁺ T cells using CRISPR gene editing. However, despite many attempts, the best we could achieve was around 30% mutation frequency, which is not good enough to carry out informative experiments. Thus, unfortunately we have not been able to repeat these experiments with AQP3 KO cells.

Figure 4. Please show primary data for CD3, ZAP70 and ERK phosphorylation as a supplementary figure. Are the MFI for P-CD3, P-ZAP70 and P-ERK normalized to total protein MFI and to cell size (FCS)? This is important because WNK1 inhibition reduces cell size as the authors showed, and this may cause an apparent reduction of P-ERK MFI, which may be an artifact.

As requested, we have added a new Supplementary Figure 5 showing example flow cytometry histograms for p-CD3 ζ , p-ZAP70 and p-ERK for the data in Figure 4A-C where T cells were stimulated with soluble anti-CD3 ϵ and anti-CD28 antibodies for 0-10 min and p-ERK for the data in Figure 4D, E, where cells are stimulated with plate bound anti-CD3 ϵ and anti-CD28 antibodies for 1 h. The data in Figure 4A-C show the mean fluorescence intensity of the phospho-proteins per cell and have not been normalized to total protein amount. The short-term (30 min) treatment of wild-type B6 T cells with the WNK1 inhibitor is unlikely to change the amount of these proteins.

Short-term (30 min) treatment with WNKi does indeed cause the cells to become around 20% smaller (see Figure 6A, C), but it is not clear to us why that should be used to normalize the flow cytometric signal for the phospho-proteins. Just because a cell is smaller doesn't mean the amount of p-ERK will necessarily be lower. We would add that in our hands FSC is not a reliable way to measure small differences in cell size and that the cells have been fixed and permeabilized prior to staining with the

phospho-specific antibodies, making size measurements even less reliable. The volume measurements in Figure 6A-C were all carried out on live unfixed cells.

I have a problem with the conclusion that the WNK1-OXSR1-STK39 pathway regulates cell volume via activation of SLC12 transporters and AQP3. This is an attractive model, which the authors show in Figure 5, but it relies on few experiments and many assumptions:

1) The conclusion that the WNK1-OXSR1-STK39 pathway regulates cell volume rests entirely on the fact that a WNK inhibitor reduces cell volume and that AQP3 inhibition impairs proliferation. How was cell volume measured? The methods section is vague about it. Because this is a central argument in the paper, this aspect needs to be investigated in more detail. Dynamic changes in cell volume can be measured, for instance, using dyes such as calcein.

In the original version of the manuscript, we presented cell volume measurements determined using a CASY counter, which uses an electrical resistance signal to determine volumes. The volume measurement is a direct numerical output from this equipment, hence the very simple description in the Methods. This data is now shown in Figure 6A.

To extend these data and we used an imaging method to obtain cell volumes. Taking T cells expressing a membrane-tethered tdTomato, we used super-resolution imaging of live cells to generate 3D images of individual cells. From this we were able to determine the cell volume. The method used is described in the Methods section (page 25, paragraph 1), and this new data is shown in a new Figure 6B-C. This confirms that TCR/CD28 stimulation causes an increase in cell size of ~11%, and that WNK1 inhibition causes the cells to shrink by ~20% and fail to increase in size in response to TCR/CD28 stimulation.

2) The model in Figure 5 shows SLC12A2 (and the introduction specifically mentions it), but there are no data to supports its role. The authors argue (page 7) that it is "impractical" to delete all four SLC12 genes that are expressed in T cells, but they have shown before that SLC12A2 is involved in T cell migration. I would therefore argue that they need to delete SLC12A2 in T cells and measure T cell function, cell cycle and RVI.

As requested, we have carried out a number of experiments to directly test the involvement of ions and SLC12A2 in T cell activation to support the model in Figure 7D (new numbering). Firstly, we showed that TCR-CD28-induced p-ERK is impaired if cells are placed in medium with reduced concentrations of both Na⁺ and Cl⁻ ions or Cl⁻ alone, showing that these ions play an important role in T cell activation. This is independent of their contribution to the osmolarity of the cell medium leaving their transport across the plasma membrane as a potentially important function. This is shown in a new Figure 5A-B.

Secondly, we showed that TCR/CD28 stimulation results in increased phosphorylation of Thr203, Thr207 and Thr212 on SLC12A, sites that are phosphorylated by OXSR1 and STK39 (new Figure 5D). Treatment with WNKi resulted in decreased p-SLC12A2, in support of the hypothesis that a WNK1-OXSR1-STK39 pathway leads to SLC12A2 phosphorylation in T cells.

Thirdly, we used bumetanide, an inhibitor of SLC12A2, and also used SLC12A2-deficient T cells to investigate the role of SLC12A2 in T cell activation. We found that inhibition of SLC12A2 or its genetic deletion did not affect anti-CD3 ϵ /anti-CD28-

induced p-ERK, or cell proliferation. The data is shown in a new Supplementary Figure 6B-E. In agreement with these observations, inhibition of SLC12A2 also had no effect on basal or TCR-induced volume (new Supplementary Figure 6F).

Our new RNAseq data in Figure 5C shows that mouse CD4⁺ T cells express 5 SLC12A-family members, several of which are known substrates of the WNK1-OXSR1-STK39 pathway. Thus, other SLC12A members may be more important than SLC12A2 in transducing WNK1 pathway signals that regulate T cell activation and proliferation. Despite their structural similarity, the family members have different, opposing functions. While WNK1-OXSR1-STK39-induced phosphorylation of SLC12A2 (NKCC1) leads to its activation and hence influx of Na⁺, K⁺ and Cl⁻ ions, phosphorylation of SLC12A6 (KCC3) or SLC12A7 (KCC4) inhibits these transporters, blocking efflux of K⁺ and Cl⁻ ions, with the net result being an increase in the intracellular concentrations of Na⁺, K⁺ and Cl⁻ ions. Bearing in mind this complication, we examined the effect of DIOA, an inhibitor of SLC12A6 and SLC12A7 (SLC12A6/7i) on T cell activation. Once again, we found that treatment of T cells with SLC12A6/7i did not affect TCR/CD28-induced p-ERK or cell division, possibly because the inhibitor is simply reinforcing their physiological inhibition by WNK1 activity downstream of the TCR. This new data is in Supplementary Figure 6B-E.

Taken together the data show that while WNK1-OXSR1-STK39 signaling appears to be active during T cell activation, resulting in phosphorylation (and presumably activation) of SLC12A2, we were unable to identify a functional requirement for SLC12A2 or SLC12A6 and SLC12A7 in TCR/CD28-induced activation. This could be because we did not mutate or inhibit the key SLC12A-family members of the 5 expressed in CD4⁺ T cells, or because other WNK1-regulated ion channels are more important. We have amended the Results, the legend to Figure 7D and the Discussion to make these points clear.

On page 12, the authors conclude that WNK1 deletion results in ATR activation independently of DNA damage. What is the evidence for that?

The evidence for this is that in WNKi-treated T cells, we see no increase in phosphorylated RPA32 (S4/8, p-RPA32) and phosphorylated H2AX (S139, γ H2AX), markers of single-stranded and double-stranded DNA damage respectively (Supplementary Figure 4C-D).

An interesting question to ask would be whether impaired RVI in WNK1 deficient T cells can be rescued (and thus impaired T cell proliferation) by blocking RVD (regulatory volume decrease) mediated by LRRC8 channels using CRISPR or DCPIB. This is not necessary but something for the authors to consider.

This is an interesting suggestion for future studies, but one that is outside the scope of the current manuscript.

Reviewer #2 (Remarks to the Author):

Comments to the Authors

This manuscript by O'may colleagues explored whether WNK1 is responsible for CD4⁺ T cell activation and T cell-dependent antibody responses. O'may et al. provide a comprehensive and detailed analysis of WNK1 downstream effector signaling pathways regulating T cell-dependent antibody responses. They found that OXSR1, STK39, and AQP3 is a downstream signaling cascade of WNK1

corresponding T cell activation. They further demonstrate that WNK1 pathway-stimulated AQP3-driven water influx is vital for TCR-mediated proliferation and cell-cycle regulation. Overall, the manuscript is well written in general, and the contents are convincing for most of the parts. However as discussed below, this study is missing several points and need to be interpreted with care.

Major Concerns:

1. In the former author's study, CCL21-induced CCR7 activation mediated cell migration to activate CD4+ T cells via the WNK1-OXSR1-STK39-SLC12A2 pathway. The current study demonstrated that the same downstream signaling cascade is critical for TCR/CD28-induced activation of CD4+ T cells independently of regulation of migration and adhesion. Although the activation of CCR7 and TCR shared their downstream effector signaling pathway, the physiological outcomes (migration vs proliferation) seem to be different. What's the reason or underlying mechanism for these differences?

This is an interesting and very broad question about signaling that has been debated for a long time, asking how it is that signaling from different receptors engages overlapping sets of signaling pathways, yet results in different physiological outcomes. The reviewer is asking specifically about the WNK1 pathway being activated by both CCR7 and TCR signaling. But the same could be asked about, for example, PI3-kinase, which is also activated by both receptors. Similarly, ERK and NF- κ B pathways are activated by many different receptors (TCR, CCR7, TNFR-family, TLR-family, etc.) which have very distinct physiological outcomes (proliferation, migration, differentiation, survival, etc.). For the most part, it is not known how this overlapping use of pathways by different receptors leads to different outcomes, but the answer is likely to lie in temporal dynamics, quantitative aspects (how much signaling of a given pathway is induced), and combinatorial effects of engaging multiple different pathways simultaneously. In addition, there may be spatial differences with engagement of the pathways in different regions or compartments of the cell. The simple answer is that we don't know how engagement of similar pathways by different receptors leads to different outcomes.

2. Authors proposed that WNK1-deficient T cells were markedly impaired in TCR/CD28-induced proliferation via the AQP3-dependent pathway. In Fig 2, the AQP3i-induced reduction of TCR/CD28-activated proliferation was less than that seen in the WNK1 deletion. This raises the question that WNK1 may regulate T cell proliferation not only via the AQP3-driven pathway but also by alternative pathways.

We agree with the reviewer that inhibition of AQP3 gives rise to a less severe defect compared to loss or inhibition of WNK1. The same is true of the OXSR1/STK39 double mutant which is also less affected compared to loss of WNK1. In both cases this suggests that WNK1 regulates T cell proliferation through proteins other than OXSR1, STK39 or AQP3. WNK1 is likely to have other substrates beyond OXSR1 and STK39, which could play important roles in T cell activation. Similarly, it is possible that other aquaporin family members may function in parallel with AQP3, since T cells also express AQP9 and AQP11. This issue is discussed on page 15, paragraph 2.

3. Current study provides evidence that the WNK1 regulates T cell proliferation and cell cycle via OXSR1/STK39/AQP3-mediated water influx. Is this an exclusive pathway or mechanism for WNK1-mediated cell-cycle regulation? OXSR1 is a well-characterized WNK1 downstream target kinase and is activated in response to osmotic stress by WNK family members. Melanie Cobb's lab (Tu

et al., PNAS 2011) reported that WNK1 regulates mitosis via the OXSR1-independent pathway. In addition, the same group recently reported WNK1 as an autophagy regulator (Gallolu Kankanamalage, Autophagy 2017; Gallolu Kankanamalage, PNAS 2016). Indeed, autophagy pathways are tightly linked with cell-cycle progression. Key experiments to exclude those alternatives would strengthen the conclusions drawn.

To strengthen the connection between WNK1 and OXSR1, we added another new experiment in Supplementary Figure 3A, showing that TCR stimulation results in increased phosphorylation of OXSR1 and that this is inhibited by WNKi, demonstrating that WNK1 kinase activity is required both for TCR-CD28-induced and basal phosphorylation of OXSR1. The experiment is referenced on page 7, paragraph 2 of the Results.

Nonetheless, as discussed above, we agree that WNK1 may well be acting via substrates other than OXSR1 and STK39. The phenotype of OXSR1/STK39 double mutant T cells is less affected compared to loss of WNK1, suggesting that WNK1 may be acting via other substrates.

The Tu et al study cited by the reviewer shows that WNK1 regulates mitosis independently of OXSR1. In this case, WNK1 could be acting via the related STK39 kinase, or indeed both OXSR1 and STK39 could be involved and functionally redundant in transducing WNK1 signals that regulate mitosis. Alternatively, the WNK1 regulation of mitosis in this study could be acting independently of both OXSR1 and STK39.

We note the studies from the Cobb group demonstrating that WNK1 regulates autophagy via STK39, and the reviewer's suggestion that this may be one route by which WNK1 regulates proliferation. This is an interesting area for future studies, but, in our view, lies outside the scope of the current study.

We address these issues in the Discussion (page 15, paragraph 2 and page 17, paragraph 3).

4. There are clear genetic links between WNK1, OXSR1, and STK39 and T cell-dependent immune response. Is there any genetic relationship between AQP3 and WNK1 or AQP3 and altered T cell-dependent immune response? Although the authors used a pharmacological AQP3 inhibitor (AQP3i) to block AQP3, the loss of function experiments such as CRISPR knockout would be required to gather convincing evidence to support their conclusions.

We are not aware of genetic linkage between WNK1 and AQP3. Our hypothesis is that WNK1 signaling via OXSR1 and STK39 and SLC12A-family ion co-transporters leads to an influx of ions which in turn causes water to passively enter the cell by osmosis through AQP3.

We agree with the reviewer that a genetic disruption of AQP3 would be a good way to extend our results with the AQP3 inhibitor. Several years ago, based on published papers, we had tried to obtain AQP3-deficient mice, but the labs we approached were unable to provide them. More recently we have identified that there is a EUCOMM allele of *Aqp3* available which can be used to generate *Aqp3* knockout mice. However, the process of importing and establishing this strain and crossing it to a germline Flp and then Cre driver to generate the knockout mice will take a substantial amount of time and is beyond the scope of what we could practically do for this response. To circumvent this, we attempted to knockout *Aqp3* in primary mouse CD4+ T cells using

CRISPR gene editing as the reviewer suggested. However, despite many attempts, the best we could achieve was around 30% mutation frequency, which is not good enough to carry out informative experiments. Thus, unfortunately we have not been able to repeat these experiments with AQP3 KO cells.

5. Authors proposed that WNK1-activated AQP3-driven water influx is the primary mechanism for CD4+ T cell proliferation mediating T cell-dependent antibody response. The authors demonstrated a mechanism that WNK1 stimulates AQP3-driven water influx through the OXSR1-STK39-SLC12A2 pathway. However, several alternative mechanisms can be proposed. It is well established that WNK1 regulates cell surface expression of multiple channels and transporters via regulating exo- and/or endocytosis of them. WNK1 can directly activate AQP3 by phosphorylation and/or physical protein interaction. Hence, it is conceivable that WNK1 may upregulate AQP3 by activating intrinsic channel properties directly (activity, etc.) or by regulating the cell-surface abundance of the AQP3. The authors need to carefully answer raised questions.

We are not aware of any evidence that WNK1 phosphorylates AQP3 or interacts with it. Since WNK1 regulates protein trafficking, it is certainly conceivable that WNK1 may regulate the trafficking of AQP3, though again there is no evidence for this. To study this requires an antibody to AQP3 that detects an extracellular epitope of AQP3 so that surface levels could be measured by flow cytometry. Since T cells are very small, with very little cytoplasm, imaging of fixed cells using immunofluorescence with antibodies to AQP3 would not reliably distinguish if AQP3 was at the surface. Unfortunately, we have not been able to identify an antibody to the extracellular domains of AQP3 that would allow us to carry out this study. We have added a comment in the discussion that WNK1 may regulate trafficking of AQP3, or indeed other proteins required for proliferation (page 17, paragraph 3).

6. It is known that AQP3 is permeable to glycerol and H₂O₂ as well as water. Glycerol and H₂O₂ uptake via AQPs are known to be required for T-cell migration and activation. The current study suggested that AQP3-driven water influx is responsible for CD4+ T cell proliferation and antibody response. Is there any chance that AQP3-mediated glycerol and/or H₂O₂ transport can contribute to regulating T cell differentiation and activation?

To address the issue of whether H₂O₂ transport may be important for TCR/CD28-induced T cell activation, we used catalase to eliminate H₂O₂. Initially we titrated the catalase and showed that 20 u/ml was sufficient to eliminate exogenously added 1 mM H₂O₂. Subsequently we incubated T cells with 20 u/ml catalase and showed that this had no effect on TCR/CD28-induced p-ERK measured after 1 h of stimulation and no effect on TCR/CD28-induced proliferation measured after 1-3 days stimulation. The same results were seen with 2 and 200 u/ml catalase. We conclude that the requirement for AQP3 activity in T cell activation is unlikely to be due to H₂O₂ transport. These data are shown in a new Supplementary Figure 7.

Regarding glycerol transport by AQP3, we note that in the in vitro assays for TCR-CD28-induced p-ERK and proliferation, there is no glycerol in the media. Thus, it is unlikely that the requirement for AQP3 activity in T cell activation is due to glycerol transport. This point is addressed on page 13, paragraph 2.

But ultimately the rescue of p-ERK signaling and G1 entry by hypotonic medium shows that it is WNK1 signaling-induced water entry that is required for these two processes.

7. In Fig. 5, hypotonic stress rescued WNK1 mutant- or WNK1i-blunted anti-CD3/CD28 response through water influx, whereas the isotonic medium with reduced Na/Cl concentrations failed to reverse it. Hence, the authors concluded that the water influx but not ion concentration is responsible for T cell activation. The authors examined isotonic medium containing low concentration NaCl. It claims several concerns. Lower external Cl⁻ concentration can shift Cl⁻ equilibrium potential in a more positive direction. It may elicit inward Cl⁻ current (Cl⁻ efflux) at RMP leading to water efflux. It is also reported that hypotonic stress promotes vascular cell proliferation via WNK1-mediated volume-regulated Cl⁻-channel (VRCC) regulation. Thus, the current study does not fully exclude the possibility that hypotonic stress regulates TD antibody response through WNK1-mediated ion channel regulation such as VRCC and Ca²⁺-activated Cl⁻ channel. Additionally, it is well established that hypertonic stress activates WNK1 phosphorylation. Does hypertonic stress mimic the TCR-induced proliferation of CD4⁺ T cells?

The reviewer suggests that isotonic medium with reduced NaCl concentration may result in Cl⁻ efflux and hence water efflux. Since this condition fails to rescue ERK activation and proliferation of WNK1-inhibited cells, if it also results in water efflux as the reviewer suggests, it would support our hypothesis that water influx is required for proliferation.

The reviewer cites work (Zhang et al 2017) in which hypotonic treatment induces proliferation of vascular smooth muscle cells by activating the volume-regulated Cl⁻-channel (VRCC) leading to Cl⁻ efflux. This decrease in Cl⁻ in turn causes increased WNK1 activation, which is required for proliferation of the cells:

hypotonic medium → VRCC activation → Cl⁻ efflux → WNK1 activation → proliferation.

While such a mechanism may also operate in T cells, it is not clear how this applies to our studies which are focused on understanding how WNK1 activation leads to proliferation, the final step in the model above. We show that when WNK1 is inhibited or deleted, T cell proliferation is decreased, and this defect can be reversed by hypotonic medium, but not by isotonic medium with a similar reduced level of ions, implying that water entry is required to overcome the WNK1 deficiency in ERK activation and proliferation. In our model:

TCR/CD28 signaling → WNK1 activation → water influx → proliferation.

In our experiments, hypotonic medium is able to bypass inhibition or deficiency of WNK1 to restore ERK activation and proliferation, and although hypotonic medium has been shown to activate WNK1 this is unlikely to play a role in our system as WNK1 activity has been blocked by inhibition of genetic deletion.

8. Deletion of one of the WNK families can lead to compensatory expression of another WNK isoform. For example, WNK1 was markedly overexpressed in WNK3 knockout mice (Mederle et al., Am J Renal Physiol 2013). Although the author's former study identified that *Wnk1* is the only member of the WNK family expressed in thymocytes, it would be important to assess whether other WNK family is upregulated in WNK1-deficient CD4⁺ cells to compensate.

The reviewer raises an important point. We addressed this by carrying out RNA sequencing on control and WNK1-deficient CD4⁺ T cells. We show that *Wnk2*, *Wnk3* and *Wnk4* are not expressed in control T cells and are not upregulated in WNK1-deficient cells. Thus, there is no compensation from other WNK isoforms and WNK1-deficient T cells lack expression of all 4 WNK kinases. These data are shown a new Supplementary Figure 1C.

9. Authors showed that WNK1 deficiency disturbed T cell proliferation and TD antibody responses. Can forced WNK1 overexpression in WNK1-deficient T cells rescue the phenotypes? Additionally, the gain of function experiment would be supporting the notion that WNK1 is required for CD4+ T cell activation.

We believe the suggested experiment of re-expressing WNK1 in WNK1-deficient T cells is not necessary. In studies using RNA interference (RNAi) to knockdown expression of target genes, it is common to re-express the target gene in order to prove that phenotypes are due to on-target effects of the RNAi. However, this is not routinely carried out in genetic knockout studies, since the alleles are well characterized and there is little doubt that the correct gene has been mutated. In our studies we have perturbed WNK1 in three distinct ways. Firstly, we analyzed T cells with a deletion of part of the *Wnk1* gene which block expression of a functional kinase. Secondly, we used mice with a point mutant allele which changes just one amino acid in WNK1 resulting in an inactive kinase. Thirdly we used a selective WNK1 inhibitor. All three approaches gave the same phenotype - decreased TCR/CD28-induced proliferation. In view of this, we believe there is little doubt that the phenotype is due to compromised WNK1 function.

10. The authors demonstrated that WNK1 kinase activity is required for the AQP3 activations using a kinase-inactive allele of WNK1 (Wnk1D368A). Apart from D368A, do other kinase active sites also affect CD4+ cell proliferation?

WNK1 has only one active site, and the D368A mutation inactivates this. The reviewer may be asking if we have mutated any other amino acids in the active site, but we have not done this, since the D368A mutation eliminates kinase activity, which was the focus of our study.

Reviewer #3 (Remarks to the Author):

In this study, Biggs O'May et al. demonstrated that water influx via AQP3 mediated by WNK1-OXSR1/STK39 signal increased the proliferation of CD4-positive T cells. Moreover, they showed that WNK1-deficient CD4-positive T cells were impaired in the differentiation into T follicular helper cells and subsequent T cell-dependent antibody responses. Regarding to the mechanism, they demonstrated that water influx mediated by the WNK1 pathway activated ERK signaling and promoted entry into G1 phase in the cell cycle. There are several suggestions for improvement.

1. The authors used WNK1^{fl/-} RCE mice to delete WNK1 specifically in CD4-positive T cells. The other immune cells including B cells should also be checked for WNK1 expression as well as CD4-positive T cells.

The ROSA26-CreERT2 allele deletes loxP-flanked alleles in all cells where ROSA26 is expressed in the mouse, which is essentially everywhere - in T cells, B cells, and all other cell types. In the experiment in Figure 1A-B, we limited the effect of the *Wnk1* deletion to T cells by making mixed bone marrow radiation chimeras in sub-lethally irradiated RAG1-deficient mice. Reconstitution of sub-lethally irradiated RAG1-deficient mice meant that only the lymphocytes were completely reconstituted by the incoming marrow cells, and mixing the test marrow with a large excess of TCR α KO marrow meant that the test marrow completely reconstituted all $\alpha\beta$ T cells, but made only a small contribution to all other hematopoietic cells, e.g. B cells. Thus, the large

majority of B cells were from the TCR α KO marrow and expressed normal levels of WNK1.

In the experiments in Figure 1C-J the WNK1-deficient OT-II T cells were transferred into WT mice, in which all cells, including B cells, still express normal levels of WNK1.

2. In Figure 1A, the chimerism of blood cells should be checked to confirm whether the proliferation of WNK1-deficient CD4-positive T cells is suppressed and whether other blood cells are affected. Is it possible to confirm the chimerism of blood cells using CD45.1 positive mice or labelling fluorescent proteins such as GFP?

As described above, in the mixed bone marrow radiation chimeras, 100% of the $\alpha\beta$ T cells have to be derived from the test marrow (*Wnk1*^{fl/+}RCE or *Wnk1*^{fl/-}RCE), since there is no other source of these cells. The other lymphocytes (B cells and $\gamma\delta$ T cells) will be a mixture of cells derived from the test marrow and the TCR α KO marrow, with a large excess (80%) of the latter. We did not measure the chimerism in these other lymphocytes in the blood or elsewhere. However, the key experiments to measuring the effects of the WNK1 mutation on CD4 T cell proliferation are those shown in Figure 1C-J where we transferred purified WNK1-deficient or control CD4+ OT-II T cells into WT recipients using a congenic marker (CD45.1 v CD45.2) to distinguish the origin of the cells. In this context all cells in the host animal express normal levels of WNK1. Here we were able to look at the proliferation of the WNK1-deficient T cells (CD45.1⁻ CD45.2⁺) in the lymph nodes of the recipient mice, in an environment where they were surrounded by cells expressing normal levels of WNK1.

3. In Figure 1A, what are the results of using only WNK1fl/- RCE mice as donors without using Tcr α -/- mice? Will there be a decrease in lymphocyte count?

Leaving out the TCR α KO marrow would mean that the test marrow (*Wnk1*^{fl/+}RCE or *Wnk1*^{fl/-}RCE) would reconstitute 100% of all lymphocyte subsets including B cells. These chimeras do not have a decrease in lymphocyte count. Just such chimeras are shown in Supplementary Figure 1A and were used as a source of WNK1-deficient CD4+ T cells which were purified from the mice. We did not use them for immune response studies, since the *Wnk1* mutation is not limited to the T cells as it is in the mixed bone marrow chimeras in Figure 1A, making any results less informative.

4. Regarding to their scheme in Figure 5, they demonstrated that SLC12A2 was involved in the mechanism. However, they didn't provide any data about SLC12A2. Phosphorylation of activation site of SLC12A2 in TCR signaling and the effect of SLC12A2 inhibition on CD4+ T cell proliferation needs to be assessed.

As requested, we have carried out a number of experiments to directly test the involvement of ions and SLC12A2 in T cell activation to support the model in Figure 7D (new numbering). Firstly, we showed that TCR-CD28-induced p-ERK is impaired if cells are placed in medium with reduced concentrations of Na⁺ and Cl⁻ ions, showing that these ions play an important role in T cell activation independently of their effects on osmolarity of the cell medium. This is shown in a new Figure 5A-B.

Secondly, we showed that TCR/CD28 stimulation results in an increased phosphorylation of SLC12A2, and that this is decreased by treatment with WNK1 inhibitor. This data is shown in a new Figure 5D.

Thirdly, we used bumetanide, an inhibitor of SLC12A2, and also used SLC12A2-deficient T cells to investigate the role of SLC12A2 in T cell activation. We found that inhibition of SLC12A2 or its genetic deletion did not affect anti-CD3 ϵ /anti-CD28-induced p-ERK, or cell proliferation. The data is shown in a new Supplementary Figure 6B-E.

Our new RNAseq data in Figure 5C shows that mouse CD4⁺ T cells express 5 SLC12A-family members, several of which are known substrates of the WNK1-OXSR1-STK39 pathway. Thus, other SLC12A members may be more important than SLC12A2 in transducing WNK1 pathway signals that regulate T cell activation and proliferation. Despite their structural similarity, the family members have different, opposing functions. While WNK1-OXSR1-STK39-induced phosphorylation of SLC12A2 (NKCC1) leads to its activation and hence influx of Na⁺, K⁺ and Cl⁻ ions, phosphorylation of SLC12A6 (KCC3) or SLC12A7 (KCC4) inhibits these transporters, blocking efflux of K⁺ and Cl⁻ ions, with the net result being an increase in the intracellular concentrations of Na⁺, K⁺ and Cl⁻ ions. Bearing in mind this complication, we examined the effect of DIOA, an inhibitor of SLC12A6 and SLC12A7 (SLC12A6/7i) on T cell activation. Once again, we found that treatment of T cells with SLC12A6/7i did not affect TCR/CD28-induced p-ERK or cell division, possibly because the inhibitor is simply reinforcing their physiological inhibition by WNK1. This new data is in Supplementary Figure 6B-E.

Taken together the data show that while WNK1-OXSR1-STK39 signaling appears to be active during T cell activation, resulting in phosphorylation (and presumably activation) of SLC12A2, we were unable to identify a functional requirement for SLC12A2 or SLC12A6 and SLC12A7 in TCR/CD28-induced activation. This could be because we did not mutate or inhibit the key SLC12A-family members of the 5 expressed in CD4⁺ T cells, or because other WNK1-regulated ion channels are more important. We have amended the Results, the legend to Figure 7D and the Discussion to make these points clear.

5. For Figure 2L-M, the authors should show the expression level of AQP family in CD4⁺ T cells or cite previous articles that show AQP3 is the predominant AQP in CD4⁺ T cells. Coupling of SLC12A2 and AQP3 shown in the summary figure 5D is not directly proved in this study. Can the effect of WNK signaling on the activity of AQP3 (such as trafficking to the cell membrane) be assessed?

As requested, we provide new RNA sequencing data in Figure 6C which shows that CD4⁺ T cells express 3 out of 11 aquaporin genes: *Aqp3*, *Aqp9* and *Aqp11*. Of these *Aqp3* is expressed at the highest level, which is why we focused our analysis on this isoform.

Our model is that WNK1 signaling leads to ion influx which in turn causes water to enter by osmosis through AQP3. It is well established that WNK1 plays an important role in the osmoregulation of cells and in modulating cell volume (de Los Heros et al 2018). In response to hypertonic stress, WNK1 is activated and transduces signals via OXSR1 and STK39 to the phosphorylation of SLC12A-family ion co-transporters such as SLC12A2 (NKCC1) leading to influx of Na⁺, K⁺ and Cl⁻ ions. This in turn results in osmotically-driven water influx which occurs passively through water channels.

Our model does not require that there be any direct coupling between AQP3 and SLC12A2 or between AQP3 and any other member of the WNK1 pathway. Nonetheless, since WNK1 has been implicated in controlling protein trafficking, WNK1 signaling may affect the trafficking of AQP3 to the plasma membrane as suggested by the reviewer. To study this requires an antibody to AQP3 that detects an extracellular

epitope of AQP3 so that surface levels could be measured by flow cytometry. Since naive mouse T cells are very small, with very little cytoplasm, imaging of fixed cells using immunofluorescence with antibodies to AQP3 would not reliably distinguish if AQP3 was at the surface. Unfortunately, we have not been able to identify an antibody to the extracellular domains of AQP3 that would allow us to carry out this study. We have added a comment in the discussion that WNK1 may regulate trafficking of AQP3, or indeed other proteins required for proliferation (page 17, paragraph 3).

6. Observation that reduced tonicity rescued WNK1 deficiency is interesting in Figure 5. Is this effect through only AQP3? Does WNK1/OXSR1/STK39 knockout or knock-in affect water permeability in T cell?

The hypotonic rescue seen in Figure 7A-C (new numbering) could involve water movement through proteins other than AQP3, e.g. AQP9 or AQP11. We do not expect mutation of the WNK1 pathway to affect water permeability of the T cells (see response to next point).

7. Water permeability should be examined, as in Fig.5 of previous paper (PMID : 22927550).

As requested, we examined water permeability of CD4⁺ T cells treated with a WNK1 inhibitor or vehicle alone to determine if inhibition of the WNK1 pathway perturbs this parameter. As a control, we also treated cells with an inhibitor of AQP3. We found that while AQP3 inhibition decreased water permeability, WNK1 inhibition had no effect (Figure R1). Water permeability was determined using a stopped flow approach with an inwardly directly osmotic gradient, recording 90° light scattering as a measure of cell shrinkage, following the method described in Figure 5 of Hara-Chikuma et al 2012 (PMID: 22927550), as suggested by the reviewer. The effect of AQP3i on water permeability has been included in the main text (new Figure 6E).

Figure R1. Inhibition of WNK1 does not affect water permeability of CD4⁺ T cells. Water permeability of CD4⁺ T cells treated with the indicated inhibitors was determined using a stopped flow approach with an inwardly directly osmotic gradient (150 mM mannitol), recording 90° light scattering as a measure of cell shrinkage. Observed rate constants from individual experiments were normalized to the average of the rate constant obtained in the absence of inhibitor (relative rates). Graph shows normalized relative rates. Each point represents the relative rate constant for each biological replicate. Mann-Whitney test. * p < 0.05; ns, not significant.

8. Intracellular localization of AQP3 should be shown, as in Fig. 2d of previous paper (PMID : 22927550). Does WNK1/OXSR1/STK39 knockout or knock-in affect intracellular localization of AQP3?

As described above, we do not expect the WNK1 pathway to affect the trafficking of AQP3. Nonetheless, since WNK1 has been implicated in controlling protein trafficking, WNK1 signaling may affect the trafficking of AQP3 to the plasma membrane. To study this requires an antibody to AQP3 that detects an extracellular epitope of AQP3 so that surface levels could be measured by flow cytometry. Since naive mouse T cells are very small, with very little cytoplasm, imaging of fixed cells using immunofluorescence with antibodies to AQP3 would not reliably distinguish if AQP3 was at the surface. Unfortunately, we have not been able to identify an antibody to the extracellular domains of AQP3 that would allow us to carry out this study. We have added a comment in the discussion that WNK1 may regulate trafficking of AQP3, or indeed other proteins required for proliferation (page 17, paragraph 3).

9. Please show cell images (micrograph) before and after cell volume is changed. Often, CASY cell counter cannot accurately measure cell volume.

In the original version of the manuscript, we presented cell volume measurements determined using a CASY counter, which uses electrical resistance to determine volumes. This data is now shown in Figure 6A. While the CASY counter gives reliable relative volumes when comparing the same cell type, the absolute numbers may not be accurate, so we have presented volumes normalized to cells that have not been stimulated or treated with inhibitor.

To obtain accurate absolute volumes we used an imaging method. Taking T cells expressing a membrane-tethered tdTomato, we used super-resolution imaging of live cells to generate 3D images of individual cells. From this we were able to determine the cell volume. Representative images of cells treated with WNKi are shown in Figure 6B and the quantitation is shown in Figure 6C. This confirms that TCR/CD28 stimulation causes an increase in cell size of ~11%, and that WNK1 inhibition causes the cells to shrink by ~20%.

10. The authors used only an AQP3 inhibitor, DFP00173, to examine the physiological role of AQP3 and did not directly evaluate water permeability of AQP3 in T cell. Importantly, DFP00173 inhibits not only water transport but also glycerol and H₂O₂ transport (PMID: 30862673). In addition, it has been reported that T cell migration is dependent on AQP3-mediated “hydrogen peroxide (H₂O₂) uptake” but not the canonical water/glycerol transport (PMID : 22927550). Therefore, there is a possibility that WNK1/OXSR1/STK39 signaling affect H₂O₂ transport, resulting in modulation of T cell migration. Further examinations of physiological analysis of AQP3 are required in this study, to answer these questions. Experiments using AQP3-deficient (knockout) T cell would be best show the direct involvement into the proliferation of CD4-positive T cells.

We agree with the reviewer that a genetic disruption of AQP3 would be a good way to extend our results with the AQP3 inhibitor. Several years ago, based on published papers, we had tried to obtain AQP3-deficient mice, but the labs we approached were unable to provide them. More recently we have identified that there is a EUCOMM allele of *Aqp3* available which can be used to generate *Aqp3* knockout mice. However, the process of importing and establishing this strain and crossing it to a germline Flp and then Cre driver to generate the knockout mice will take a substantial amount of time and is beyond the scope of what we could practically do for this response. To circumvent this, we attempted to knockout *Aqp3* in primary mouse CD4+ T cells using

CRISPR gene editing. However, despite many attempts, the best we could achieve was around 30% mutation frequency, which is not good enough to carry out informative experiments. Thus, unfortunately we have not been able to repeat these experiments with AQP3 KO cells.

To address the issue of whether H₂O₂ transport may be important for TCR/CD28-induced T cell activation, we used catalase to eliminate H₂O₂. Initially we titrated the catalase and showed that 20 u/ml was sufficient to eliminate exogenously added 1 mM H₂O₂. Subsequently we incubated T cells with 20 u/ml catalase and showed that this had no effect on TCR/CD28-induced p-ERK measured after 1 h of stimulation and no effect on TCR/CD28-induced proliferation measured after 1-3 days stimulation. The same results were seen with 2 and 200 u/ml catalase. We conclude that the requirement for AQP3 activity in T cell activation is unlikely to be due to H₂O₂ transport. These data are shown in a new Supplementary Figure 7.

Regarding glycerol transport by AQP3, we note that in the in vitro assays for TCR-CD28-induced p-ERK and proliferation, there is no glycerol in the media. Thus, it is unlikely that the requirement for AQP3 activity in T cell activation is due to glycerol transport. This point is addressed on page 13, paragraph 2.

But ultimately the rescue of p-ERK signaling and G1 entry by hypotonic medium shows that it is WNK1 signaling-induced water entry that is required for these two processes.

Minor

1. There are many abbreviations (ex. RCE) that are not spelled out in the manuscript.

The RCE abbreviation had been previously defined in the Methods. We have now edited the Results (page 4, last paragraph) to make the meaning of this abbreviation clear (it refers to the ROSA26-CreERT2 allele). In addition, the full names for all the inhibitors (WNKi, AQP3i, ATRi, SRCi) have been added to the Figure legends where any of these inhibitors is used, to make it easy for the reader to know which chemical was used.

2. The authors group revealed previously that WNK1 is involved in attachment and migration of CD4+ T cells (Köchler et al. Nat Immunol 2016) and recently showed the same molecular mechanism is important in migration and proliferation of B cells (Hayward et al. 2021 <https://doi.org/10.1101/2021.09.09.459588>). It was also reported previously that AQP3 is involved in keratinocyte's proliferation and migration (Hara-Chikuma et al. J Mol Med 2008), and T cell migration (Hara-Chikuma et al. J Exp Med 2012). Based on these publications, involvement of WNK1 signaling and AQP3 in CD4 T cell proliferation is interesting but could have less impact. The authors must emphasize the novelty of this manuscript.

The current study has many novel aspects. We show for the first time that WNK1 is required for TD antibody responses and for antigen-driven proliferation of CD4 T cells in vivo and in vitro. We show for the first time that WNK1, OXSR1, STK39 and AQP3 are all required for TCR/CD28-induced proliferation in vitro. Moreover, we provide mechanistic insights, showing that the WNK1 pathway regulates signaling from the TCR and CD28 to p-ZAP70 and p-ERK, is required for commitment from G0 to G1, for progression through S phase and to suppress the ATR-dependent checkpoint in G2. None of this has been previously reported. Most importantly, we show for the first time that forcing water into T cells with hypotonic medium rescues the defect in TCR-induced p-ERK induction and Ki67 upregulation in WNKi-inhibited cells.

3. The authors concluded in the previous paper (Köchli et al. Nat Immunol 2016) that WNK1 does not contribute to ERK activation in TCR signaling. This is inconsistent with Figure 4 in the current manuscript and needs to be discussed.

The reviewer is referring to Supplementary Figure 3b in our earlier paper (Köchli et al Nat Immunol 2016). The experiment shown there was analysis of Jurkat cells, a human T cell leukemia line, stimulated with an anti-CD3 antibody, in which WNK1 had been knocked down by RNA interference. The experiments in the current study used primary naive mouse CD4+ T cells which either had a genetic mutation eliminating WNK1 kinase activity, or were treated with a WNK inhibitor. Thus, there are many differences between these experiments which could account for the difference in observed outcome. Primary mouse T cells may have different signaling pathways compared to an immortalized leukemic cell line. RNA interference is a technique which gives partial knockdown of target genes. With WNK1 we achieved a knockdown of 60% in Jurkat cells (see Supplementary Figure 1a in the Köchli et al 2016 paper), leaving 40% of the mRNA remaining. This may not be enough to reveal the ERK signaling defect. In contrast, the genetic deletion of *Wnk1* in the mouse system is likely to be close to 100% (see Supplementary Figure 1B of the current manuscript), and treatment with WNKi is likely to inhibit almost all kinase activity. Any or all of these differences could account for the different result seen in the mouse system.

Response to Reviewer's Comments

Reviewer #1 (Remarks to the Author):

The authors have addressed many of the previously raised concerns by conducting a number of new experiments and rewriting some sections of the manuscript. These changes have clarified some of the issues that were brought up and improved the study. Nonetheless, I am not entirely convinced of some of their conclusions and the proposed model in Figure 7D, which is not fully supported by the data.

1. The new experiments in which SLC12A2 was inhibited or deleted in T cells do not provide any evidence that SLC12A2 (or another homologue of the SLC12 family) regulates ERK phosphorylation, cell volume or water influx. The fact that SLC12A2 is phosphorylated in a WNK1 dependent manner is not sufficient to implicate it in T cell function. To be fair, the authors appropriately discuss these limitations in the text, but should also remove SLC12A2 from the model in Figure 7.

We have removed 'SLC12A2' from Figure 7d and replaced it with '?' to indicate that we do not know the identity of the ion transporters.

2. The experiments with the WNK inhibitor in Figure 3 show reduced ZAP70 and ERK phosphorylation after TCR stimulation. A much weaker reduction of ERK phosphorylation is observed with the AQP3 inhibitor. These data suggest that WNK1 likely has direct effects on TCR signaling that are independent of AQP3, which may explain much of the defects in T cell proliferation, cell cycle arrest and in vivo phenotype (Tfh cells etc).

We agree with the Reviewer that inhibiting WNK1 has a stronger effect on ERK phosphorylation than inhibiting AQP3. Thus, WNK1 is signaling via pathways that are independent of AQP3. We discuss this point in the Discussion (page 16, paragraph 3). This stronger effect of WNK1 inhibition may be caused by redundancy between AQP3 and the other aquaporins expressed in naïve CD4+ T cells (Figure 6d), i.e. WNK1 activity may induce water influx through AQP9 and AQP11 to drive proliferation in the absence of AQP3 activity. However, we fully accept that WNK1 likely signals through other AQP-independent pathways to drive T cell proliferation. Some possibilities are discussed on page 19, paragraph 4, referencing published studies which show that WNK1 signals through an array of different proteins, including AMPK and PI3K.

Nonetheless, we point out that we rescue the WNK1-deficient TCR-induced pERK and G1 entry phenotypes using hypotonic media showing that water influx is the primary function of WNK1 required for these processes. In combination with the molecular components downstream of WNK1 that we have elucidated, this provides an important contribution to our understanding of the role of WNK1 in T cell activation.

3. Because of these concerns and the fact that water influx through AQP3 is a central conclusion of their study, some form of genetic deletion (CRISPR, RNAi, KO) to validate these findings is important in my opinion. As pointed out in the original critiques, the involvement of AQP3 in T cell function is solely based on the use of an inhibitor. This limitation is made worse by the fact that the

molecular link between WNK1 and AQP3 is missing (as SLC12A2 has been ruled out).

As requested, we have used a genetic disruption of *Aqp3* to study its function. We imported ES cells with the *Aqp3*^{tm2a(EUCOMM)Wtsi} (*Aqp3*^{tm2a}) allele and established mice bearing this allele. This EUCOMM allele has a LacZ and a neo gene and a splice acceptor site and 2 polyadenylation sites integrated in intron 1 of the *Aqp3* gene (Supplementary Figure 3d). Usually, these EUCOMM alleles result in loss of function, and that is what we observed. Q-PCR shows a loss of 95-98% of *Aqp3* mRNA in homozygous *Aqp3*^{tm2a/tm2a} CD4⁺ T cells (Supplementary Figure 3e).

The results show that AQP3-deficient T cells have reduced TCR/CD28-induced p-ERK and reduced upregulation of Ki67, a marker of entry into G1, although there was no change in cell division (Figure 6h-k). The phenotype of the AQP3-deficient T cells was milder than that seen with the AQP3 inhibitor. This difference could be due to a number of causes. There may be residual expression of AQP3 in the genetic mutant facilitating flux through the TCR-ERK signaling pathway and normal T cell proliferation. Alternatively, because the *Aqp3*^{tm2a/tm2a} cells experience a constitutive loss of AQP3 activity, they may counteract it through compensatory changes to the expression of other proteins. The inhibitor may also act on the other AQP proteins or have other off-target effects. Future work will be needed to distinguish these possibilities. Nonetheless, our results show that water entry is required for TCR/CD28 induced p-ERK and Ki67 upregulation and that water is likely to enter in part through AQP3. These points are discussed on page 18, paragraph 2.

4. From the experiments in Figure 5, in which NaCl concentrations in the media are reduced, the authors conclude “Na⁺ and Cl⁻ ions are likely important for TCR/CD28 signaling” because ERK phosphorylation is reduced. These experiments are used to support the idea that Na⁺ and Cl⁻ influx by SLC12A promotes subsequent water influx by AQP3, but the evidence is weak. In addition to the concerns discussed above, I see two additional problems:

A) the authors do not show that Na⁺ and/or Cl⁻ uptake is indeed altered in *Wnk1*, *Oxsr1* or *Stk39* deficient T cells to support the idea that *Wnk1* regulates Na⁺, K⁺ or Cl⁻ levels in cells. Na⁺ levels can be measured, for instance, with fluorescent Na⁺ dyes; K⁺ can be measured by thallium flux measurements. These experiments are absolutely critical in my opinion.

We agree that this is an important experiment. As requested by the Reviewer, we have carried out K⁺ flux measurements using the thallium method suggested. Thallium ions are used in place of K⁺ by ion transporters and their influx can be monitored using a fluorescent dye. This analysis showed that TCR/CD28 stimulation increases the rate of K⁺ influx as seen by thallium entry and that this is blocked by the WNK inhibitor (Figure 5a). We conclude that TCR-CD28 stimulation leads to a WNK1-dependent K⁺ influx.

B) Na⁺ and Cl⁻ ions may affect TCR signaling by altering the membrane potential. A negative membrane potential is essential for T cell signaling as many studies in K⁺ and Na⁺ channel deficient T cells have established. If the authors want to use the data in Figure 5A,B to support their model, they need to exclude effects on the membrane potential as a reason for the observed decrease in ERK phosphorylation.

We agree that changes to extracellular ion concentrations could affect T cell signaling by changing the membrane potential. As requested by the Reviewer, we have

measured membrane potential in CD4+ T cells using fluorescence. We found that in isotonic medium in which NaCl had been replaced by L-glucose or by Na gluconate or in hypotonic medium, the membrane potential was either slightly changed or not changed at all (Supplementary Figure 7a, b). Thus, decreased p-ERK activation seen when NaCl was reduced (Figure 5b, c) or the rescued p-ERK signal in hypotonic medium (Figure 7a, b) are unlikely to be caused by altered membrane potential.

Reviewer #2 (Remarks to the Author):

Comments to the Authors

Most comments in this revised manuscript have been addressed nicely with additional data to support their conclusions. However, several concerns still need clarification for the manuscript to be considered credible and improved.

1. In response to comment #7 from reviewer 2, my questions were regarding two aspects: 1) The effects of lowering extracellular Cl⁻ shifting its equilibrium potential eliciting inward Cl⁻ currents on WNK1 activation and its downstream events.

The Reviewer is referring to our experiments in which hypotonic medium with reduced levels of Na and Cl rescued p-ERK and Ki67 upregulation in WNK1-deficient or WNK1 inhibited cells. In their comments on the original manuscript, the Reviewer suggested that the reduced extracellular Cl⁻ in the hypotonic medium would result in Cl⁻ efflux (inward Cl⁻ current), which in turn would cause WNK1 activation since Cl⁻ is an inhibitor of WNK1 and that this WNK1 activation would be contributing to downstream events such as p-ERK and cell proliferation. We believe the Reviewer is referring to a study by Zhang et al (2018) (PMID:28770829). Specifically, the Reviewer previously suggested that:

'hypotonic stress regulates TD antibody response through WNK1-mediated ion channel regulation such as VRCC and Ca²⁺-activated Cl⁻ channel'.

We thank the Reviewer for bringing this possibility to our attention.

The Zhang et al study showed that hypotonic treatment induces proliferation of vascular smooth muscle cells by activating the volume-regulated Cl⁻ channel (VRCC) leading to Cl⁻ efflux. This decrease in Cl⁻ in turn causes increased WNK1 activation, which is required for proliferation of the cells. The causal sequence of events in this published study is:

hypotonic medium → VRCC activation → Cl⁻ efflux → WNK1 activation → proliferation.

While such a mechanism may exist in T cells, we do not believe it is directly relevant to our study which examines events downstream of WNK1, not upstream of it. Specifically, we investigated why WNK1 activation was required for proliferation, the final step of the model above. We show that hypotonic NaCl-low medium restores proliferation in TCR/CD28-stimulated T cells in which we had mutated WNK1 or inhibited it. Since this rescue does not happen with isotonic NaCl-low medium, we conclude that water entry, and not changes to [Cl⁻], can substitute for WNK1 activity and drive ERK activation and proliferation in WNK1-deficient cells. Therefore, our model is:

TCR/CD28 signaling → WNK1 activation → water influx → ERK → proliferation.

Importantly, even if the hypotonic NaCl-low medium results in VRCC-mediated Cl⁻ efflux, any potential effect of Cl⁻ efflux on WNK1 signaling in the WNK1-treated or

Wnk1^{D368A/-} cells is not likely to be relevant since there is no active WNK1 kinase in these cells. Nonetheless, it is possible that Cl⁻ efflux induced by the hypotonic medium may contribute to proliferation by another mechanism that does not involve WNK1. We have added a discussion of this point on page 17, paragraph 2. In addition, we have referred to the proliferation of vascular smooth muscle cells induced by hypotonic medium via WNK1 in the Discussion on page 20, paragraph 1.

2) The effects of hypertonic stress on WNK1 phosphorylation and the subsequent WNK1-induced events. The authors must provide thorough answers to these raised questions for a more comprehensive understanding of their findings.

This question relates to the Reviewer's comment #7 on the original manuscript, where the Reviewer asked: 'Does hypertonic stress mimic the TCR-induced proliferation of CD4+ T cells?'

As requested, we have directly tested this idea by placing CD4+ T cells in hypertonic medium. As expected, this resulted in the activation of WNK1, but did not induce p-ERK or cell proliferation (Supplementary Figure 6a-d). Thus, while WNK1 activity is required for TCR/CD28-induced p-ERK and cell proliferation, it is not sufficient. This is not a surprising result, since TCR/CD28 signaling activates many different pathways (Ca²⁺ flux, PI3-kinase, NF-κB, RAS/MEK/ERK, etc) in addition to WNK1. Full T cell activation is likely to result from the combined output of several pathways. Our results show that on its own, WNK1 activation is not sufficient for activation and proliferation of T cells.

2. A key finding of this study is that TCR activation leads to water entry, which regulates CD4+ T cell proliferation and T-cell-dependent antibody responses through a WNK1 kinase activity-dependent mechanism. While the authors have demonstrated TCR-stimulated ERK phosphorylation and conducted a *Wnk1*(D368A) knock-in experiment, they have not examined WNK1 phosphorylation and expression levels. It is imperative that the authors provide this data to strengthen the evidence supporting their conclusions.

We agree with the Reviewer that it is important to measure the expression of the WNK1-D368A mutant, and to show that this mutation blocks WNK1 activity. As requested, we have added new Q-PCR data showing that the mRNA expression of the *Wnk1*-D368A allele is the same as a wild-type allele (Supplementary Figure 2c), i.e. the point mutation does not alter expression.

Regarding WNK1 activity, the best measure of this is to quantitate p-OXSR1 (S325), a well-described direct WNK1 substrate. Importantly, we published that the WNK1-D368A mutant blocks TCR-induced pOXSR1 (Figure 5d of Köchl et al, Nat Immunol 2016 - see figure panel below). Therefore, as expected, the D368A mutation eliminates TCR-induced WNK1 activity. We have referenced these points clearly in the manuscript on page 6, paragraph 2.

3. The authors conducted new RNA sequencing analysis in *Wnk1*^{+/-}-RCE and *Wnk1*^{-/-}-RCE CD4⁺ T cells and identified three detectable isoforms of AQP genes in CD4⁺ T cells through RNASeq analysis. Have any differences been observed in the expression levels of WNK1 downstream targets, including AQPs, OXSRI, STK39, and NKCCs, by *Wnk1*^{-/-}?

As requested, we have added graphs derived from the RNAseq data showing expression of the *Oxsr1*, *Stk39*, *Slc12a*-family and *Aqp*-family genes in WNK1-deficient T cells, compared to control cells (Supplementary Figure 1d-f). These show some small changes. WNK1-deficient cells had increased expression of *Oxsr1*, *Stk39*, *Slc12a2*, *Slc12a4* and *Aqp9* and decreased expression of *Slc12a6*, *Slc12a7* and *Aqp3*.

In addition, we have added a new Supplementary Table 2 in which we give the expression values of all genes in all 12 samples (6 of each genotype). Furthermore, the raw data has been deposited in GEO and will of course be freely available for anyone to carry out their own analysis.

4. The authors proposed that Na⁺ and Cl⁻ ions play a significant role in TCR/CD28 signaling, independently of their effect on osmolarity. To enhance the understanding of this mechanism, the authors should elaborate on the underlying mechanism behind the role of Na⁺ and Cl⁻ ions on T cell signaling and WNK1 regulation. Providing this information will help establish a more comprehensive framework for their findings.

The Reviewer is referring to the data in Figure 5b, c which shows that lowering Na and Cl extracellular concentrations reduces TCR/CD28-induced p-ERK signaling independently of changes to extracellular osmolarity as total medium osmolarity is maintained in this experiment by replacing Na⁺ and Cl⁻ with L-glucose or gluconate.

This evidence aligns with our model for a role for WNK1-dependent water influx in TCR-induced p-ERK activation. In our model, TCR-induced WNK1 signaling results in the influx of Na⁺ and Cl⁻ ions resulting in subsequent movement of water into the cell via osmosis. The influx of water into the cell rebalances the intracellular (and extracellular) osmolarity to its baseline level. Thus, extracellular and intracellular tonicity are unchanged by WNK1 activity. We show that this WNK1-regulated water influx is required for TCR-induced p-ERK and proliferation. In our hypothesis, the importance of Na and Cl ions in TCR signaling is that their WNK1-regulated influx is required to cause water entry, which in turn is required for TCR/CD28-induced p-ERK and proliferation.

It is important to note that this function for WNK1 is distinct from its role in osmoregulation, where it plays an important role in regulatory volume increase (RVI).

Treatment of cells with hypertonic medium results in cell shrinkage as water leaves, causing WNK1 activation and hence ion influx and subsequent water entry by osmosis and re-swelling of the cell.

Nonetheless, it is certainly possible that Na⁺ and Cl⁻ ions may have other roles in T cells beyond the one we describe in this manuscript. We believe that is outside the scope of the current work. We have added this point to the Discussion on page 17, paragraph 2.

Overall, addressing these concerns and providing the necessary data and explanations will substantially improve the credibility and impact of the manuscript.

Reviewer #3 (Remarks to the Author):

Although inadequate in some areas, many parts of the system answer my request. However, some major comments need to be answered. In particular, I would like you to demonstrate that inhibition of WNK1 suppresses water permeability in this system.

Major:

1: Water permeability of AQPs is usually measured by stopped flow approach with osmotic gradient, since AQPs are channels but not transporters. Osmotic gradient is required to induce water transport through AQPs. In this study, the authors concluded that TCR/CD28 stimulation activated SLC12A-family of ion co-transporters, leading to the net influx of Na⁺, K⁺ and Cl⁻ ions. Inward ion flux worked as osmolytes which was driving force for increasing water permeability via AQP3. However, in Figure R1, osmotic gradient was formed by inward 150 mM mannitol, which overwhelmed small osmotic gradient formed by WNK1-OXSR1-STK39 signaling; therefore, inhibition of WNK1 did not affect water permeability. Instead of mannitol, the authors should use isotonic buffer containing Na⁺, K⁺ and Cl⁻ ions to evaluate WNK1-induced AQP3 water transport after TCR/CD28 stimulation. Furthermore, the authors should simply prove that inhibition of WNK1 reduces water permeability in this T cell system.

It is our hypothesis that TCR/CD28 stimulation results in a WNK1-dependent water flux, rather than a change in water permeability. From the Reviewer's comments we believe that they are asking us to investigate this water flux.

As previously requested, to measure water permeability of the T cells, we used the stopped flow method that the Reviewer had pointed us to in Figure 5 of a paper by Hara-Chikuma et al 2012 (PMID:22927550). Water permeability of the plasma membrane is defined as the volume of water that passes through a membrane as a function of time and applied pressure. In this method, cells are pushed into a chamber where they are mixed with hypertonic medium, resulting in water efflux and cell shrinkage. This change in volume is followed indirectly using side scatter, which increases as the cells shrink in volume. The rate at which side scatter increases (i.e. the rate of cell shrinkage) is a measure of how permeable the cells are at a given osmotic pressure. As requested in the Reviewer's original comments we used this method to analyze water permeability of the T cells, showing that, as expected, inhibition of AQP3 reduced permeability (Figure 6e). In contrast, inhibition of WNK1 did not change permeability. This is also a result we would have expected, since we have no reason to think that WNK1 directly affects water permeability. These data were presented to the Reviewers in our previous response.

However, we propose that WNK1 is involved in TCR/CD28-induced water flux - defined as the volume of water that crosses the membrane - into the cell under isotonic conditions where no external osmotic pressure is applied. We measured this by monitoring cell volume, using two methods. We used the CASY cell counter and separately we used super-resolution imaging to directly determine cell volume. These studies clearly showed that TCR/CD28 stimulation results in an increased cell volume and that this increase is dependent on WNK1 (Figure 6a-c). In addition, we note that measuring cell volume by imaging is a more direct method than using side scatter as a proxy for volume as used in the stopped flow method. These data support our hypothesis that TCR/CD28 signaling causes WNK1-dependent water influx into the cell and, we believe, directly answers the Reviewer's question.

2: In this revision, the authors have additionally performed RNA-seq using WNK1+/- RCE and WNK1-/- RCE CD4+ T cells. Analyzing these data, can WNK1-/- have suppressed any transcriptional signatures associated with TCR signaling and/or ATR-mediated G2/M checkpoint?

As requested by the Reviewer we have carried out pathway analysis on the RNAseq data using Gene Set Enrichment Analysis (GSEA) to compare Wnk1+/-RCE and Wnk1-/-RCE CD4+ T cells, looking at TCR signaling and ATR-G2/M pathways. Here are the results:

From the KEGG collection of pathways: the transcriptional signature associated with TCR signaling is reduced in WNK1-deficient T cells with a normalised enrichment score (NES) of -1.89, although the FDR q-value is 0.065 so this change is not statistically significant (using the significance threshold of $q = 0.05$).

From the Reactome collection of pathways: the transcriptional signature associated with TCR signaling is reduced in Wnk1-deficient T cells with a normalised enrichment score (NES) of -1.63, although the FDR q-value is 0.151 so this change is not statistically significant (using the significance threshold of $q = 0.05$).

From the Reactome collection of pathways: the transcriptional signature associated with G2/M checkpoints is increased in Wnk1-deficient T cells with a normalised enrichment score (NES) of 2.23 and FDR q-value of 0.042. This increase is statistically significant (using the significance threshold of $q = 0.05$).

Overall, this shows no change in TCR signaling pathways, and a small increase in the G2/M checkpoints pathway in WNK1-deficient T cells. We do not think the results from this GSEA are sufficient evidence to conclude that WNK1 regulates ATR signalling flux by modulating expression of components at the transcriptional level. Therefore, we

propose not to include it in the manuscript, unless the Reviewer and/or Editor thinks otherwise.

However, we have added a new Supplementary Table 2 in which we give the expression values of all genes in all 12 samples (6 of each genotype). In addition, the raw data has been deposited in GEO and will of course be freely available for anyone to carry out their own analysis.

We note that the effect of the WNK inhibitor is very similar to that of the WNK1 knockout. With respect to the analysis of TCR/CD28-induced p-ERK, since the cells are only activated in the presence of the inhibitor for 1 hour, it is unlikely that gene expression will have changed substantially and the observed impaired induction of p-ERK is most likely due to an acute requirement for WNK1 kinase activity during the stimulation and not to gene expression changes.